# Environmental Drivers of Circum-Antarctic Glacier and Ice Shelf Front Retreat over the Last Two Decades

Celia A. Baumhoer[1], Andreas. J. Dietz[1], Christof Kneisel[2], Heiko Paeth[2] and Claudia Kuenzer[1,2]

[1]German Remote Sensing Data Center (DFD), German Aerospace Center (DLR), Muenchner Strasse 20, D-82234 Wessling, Germany

[2]Institute of Geography and Geology, University Wuerzburg, Am Hubland, D-97074 Wuerzburg, Germany

*Correspondence to*: Celia A. Baumhoer (celia.baumhoer@dlr.de)

**Abstract.** The safety band of Antarctica, consisting of floating glacier tongues and ice shelves, buttresses ice discharge of the Antarctic Ice Sheet. Recent disintegration events of ice shelves along with glacier retreat indicate a weakening of this important safety band. Predicting calving front retreat is a real challenge due to complex ice dynamics in a data-scarce environment that are unique for each ice shelf and glacier. We explore to what extent easy to access remote sensing and modelling data can help to define environmental conditions leading to calving front retreat. For the first time, we present a circum-Antarctic record of glacier and ice shelf front change over the last two decades in combination with environmental variables such as air temperature, sea ice days, snowmelt, sea surface temperature and wind direction. We find that the Antarctic Ice Sheet area decreased by $-29,618\pm1,193$ km$^2$ in extent between 1997-2008 and gained an area of $7,108\pm1,029$km$^2$ between 2009 and 2018. Retreat concentrated along the Antarctic Peninsula and West Antarctica including the biggest ice shelves (Ross and Ronne). In several cases, glacier and ice shelf retreat occurred in conjunction with one or several changes in environmental variables. Decreasing sea ice days, intense snowmelt, weakening easterlies and relative changes in sea surface temperature were identified as enabling factors for retreat. In contrast, relative increases in mean air temperature did not correlate with calving front retreat. For future studies a more appropriate measure for atmospheric forcing should be considered, including above zero-degree days and temperature extreme events. To better understand drivers of glacier and ice shelf retreat it is critical to analyze the magnitude of basal melt through the intrusion of warm Circumpolar Deep Water that is driven by strengthening westerlies, and to further assess surface hydrology processes such as meltwater ponding, runoff and lake drainage.

## 1    Introduction

A safety band of floating ice shelves and glacier tongues fringes the Antarctic Ice Sheet (Fürst et al., 2016). Large glacier tongues and ice shelves create buttressing effects, decreasing ice flow velocities and ice discharge (De Rydt et al., 2015; Gagliardini et al., 2010; Royston and Gudmundsson, 2016). The recent large-scale retreat of ice shelf and glacier fronts along the Antarctic Peninsula (AP) and the West Antarctic Ice Sheet (WAIS) indicates a weakening of this safety band (Rott et al., 2011; Rankl et al., 2017; Friedl et al., 2018; Cook and Vaughan, 2010). Calving front retreat can increase ice discharge and hence the contribution to global sea level rise (De Angelis and Skvarca, 2003; Seehaus et al., 2015). Increases in ice discharge

occur if ice shelf areas with strong buttressing forces are lost (Fürst et al., 2016).The current contribution of the Antarctic Ice Sheet to global sea level rise is 7.6± 3.9 mm (1992-2017) but over this study period a strong trend of mass loss acceleration was observed for West Antarctica after ice shelves and glaciers retreated and thinned (IMBIE, 2018). In contrast, there is no clear trend in the mass balance of the East Antarctic Ice Sheet (EAIS). Since the 1990s altimetry measurements have shown a

small gain (but with high uncertainties) for the EAIS with 5±46 Gt/yr (1992-2017) (IMBIE, 2018). However, a strong mass loss trend of -47±13 Gt/yr (1989-2017) is calculated using the mass budget method (Rignot et al., 2019). Glacier terminus positions along the EAIS experienced a phase of retreat between 1974 and 1990 followed by a phase of advance until 2012. The single exception to this advance in East Antarctica is Wilkes Land, where retreating glacier fronts were observed (Miles et al., 2016).

The coastline of the Antarctic Ice Sheet is defined as the border between the ice sheet and the ocean (Liu and Jezek, 2004), extending along glacier and ice shelf fronts. Throughout this paper, we refer to floating glacier tongues and ice shelves when using the term "glacier and ice shelf front" as well as "calving front". Whether a glacier or ice shelf front advances or retreats depends mainly on four different factors: internal ice dynamics, geometry, external mechanical forcing and external environmental forcing (Alley et al., 2008; Benn et al., 2007; Cook et al., 2016; Luckman et al., 2015; Walker et al., 2013). The

combined influence of those factors make it challenging to create a realistic calving law and making iceberg calving still one of the least understood ice shelf processes (Bassis, 2011). Frontal retreat starts with the formation of a crevasse originating from a strain rate surpassing the yield stress of ice (Mosbeux et al., 2020). For ice shelves and floating glacier tongues, the calving position evolves where crevasses develop into through-cutting fractures (rifts). These ice shelf rifts can propagate further into the ice front or intersect with other rifts resulting into a tabular iceberg calving event (Benn et al., 2007; Joughin

and MacAyeal, 2005), where the extent of the ice shelf and the size of the iceberg is defined by the rift location (Lipovsky, 2020; Mosbeux et al., 2020; Walker et al., 2013). Further boundary conditions such as fjord geometry (Alley et al., 2008; Catania et al., 2018), ice rises and rumples (Matsuoka et al., 2015), and bed topography (Hughes, 1981) influence the stability of the calving margin.

The previously described natural cycle of growth and decay of a glacier or ice shelf (Hogg and Gudmundsson, 2017) can be

disturbed by external mechanical and environmental forces. It was found that mechanical forcing arises from tides (Rosier et al., 2014), ocean swell (Massom et al., 2018), iceberg collision (Massom et al., 2015) and tsunamis (Walker et al., 2013), which can destabilize floating ice shelves and glacier tongues initiating iceberg calving (Alley et al., 2008). Glaciers and ice shelves are in direct interaction with the atmosphere and ocean and are hence sensitive to changes in environmental conditions (Vaughan and Doake, 1996; Kim et al., 2001; Domack et al., 2005; Wouters et al., 2015). Environmental drivers can destabilize

floating ice tongues and shelves through various oceanic and atmospheric forced processes causing mechanical weakening and fracturing (Pattyn et al., 2018). Instead of sporadic calving by tabular icebergs (a sign of a natural calving event within the calving cycle) more frequent small-scale calving events can occur as a result of environmental forcing (Liu et al., 2015). Destabilization through atmospheric forcing arises from thinning through surface melt (Howat et al., 2008), hydrofracture (Kopp et al., 2017; Pollard et al., 2015), lake ponding and lake drainage (Banwell et al., 2013; Leeson et al., 2020). Ocean

forcing weakens the floating ice as a result of basal melt inducing thinning and grounding line retreat (Konrad et al., 2018; Paolo et al., 2015; Rignot et al., 2013), as well as by warmer ocean surface water undercutting the ice cliff at the waterline (Benn et al., 2007) by reducing stabilizing fast ice (Larour, 2004) and through enhanced sea ice reduction (Massom et al., 2018; Miles et al., 2016). Environmental forcing on tidewater glaciers has been frequently observed on Greenlandic glaciers (Cowton et al., 2018; Howat et al., 2008; Luckman et al., 2015) but for Antarctica in many regions it is unclear where and at

what amount environmental drivers cause calving front retreat (Baumhoer et al., 2018; Pattyn et al., 2018).

Current knowledge on environmentally-forced calving front retreat in Antarctica can be summarized by the following studies. Glacier retreat along the Antarctic Peninsula was first only associated with atmospheric warming (Cook et al., 2016; Mercer, 1978), until more recent studies identified ocean forcing as the main driver (Cook et al., 2016; Wouters et al., 2015). Additionally, the formation of melt ponds on the ice shelf surface has been discussed as an enhancing factor for calving.

Meltwater can initiate crevasse propagation resulting in hydrofracture and ice shelf retreat (Scambos et al 2000, Scambos 2017). The poleward shift of the west wind drift causes upwelling Circumpolar Deep Water (CDW). This allows warmer ocean waters to reach  the bottom of ice shelves, inducing basal melt and ice shelf thinning, as observed in the Bellingshausen and Amundsen Sea Sector (Dutrieux et al., 2014; Thoma et al., 2008; Wouters et al., 2015). Basal melt, when combined with  a retrograde bed (Scheuchl et al., 2016; Hughes, 1981), has led to a retreat of the grounding line followed by increased ice

discharge (Konrad et al., 2018; Rignot et al., 2013), for example at the former Wordie Ice Shelf (AP) (Friedl et al., 2018; Walker and Gardner, 2017). In contrast, the retreat of calving fronts along Wilkes Land (EAIS) has been associated with a reduction in the duration of the sea ice cover (Miles et al., 2016).

The drivers of ice shelf and glacier retreat and advance can be manifold depending on studied variables, time periods and regions. Therefore, the identification of environmental driving forces for fluctuations in ice shelf and glacier extent is

challenging and has been subject of many discussions in the past. So far, there has not been a comprehensive analysis comparing circum-Antarctic glacier terminus change on a continental scale within uniform time scales (Baumhoer et al., 2018). In this paper, we explore Antarctic calving front change over the last two decades by analysing Antarctic coastal change (see Figure 1) and address the question whether or not environmental drivers forced the observed changes in glacier and ice shelf front fluctuations. We compare changes in Antarctic calving fronts in two decadal time steps (1997-2008 and 2009-2018) to

minimize the effect of short-term glacier front fluctuations. To identify potential links between calving front retreat and recent changes in the Antarctic environmental conditions, we correlated two decades of glacier change with climate data, including air and sea surface temperature as well as changes in wind direction, snowmelt and sea ice cover.

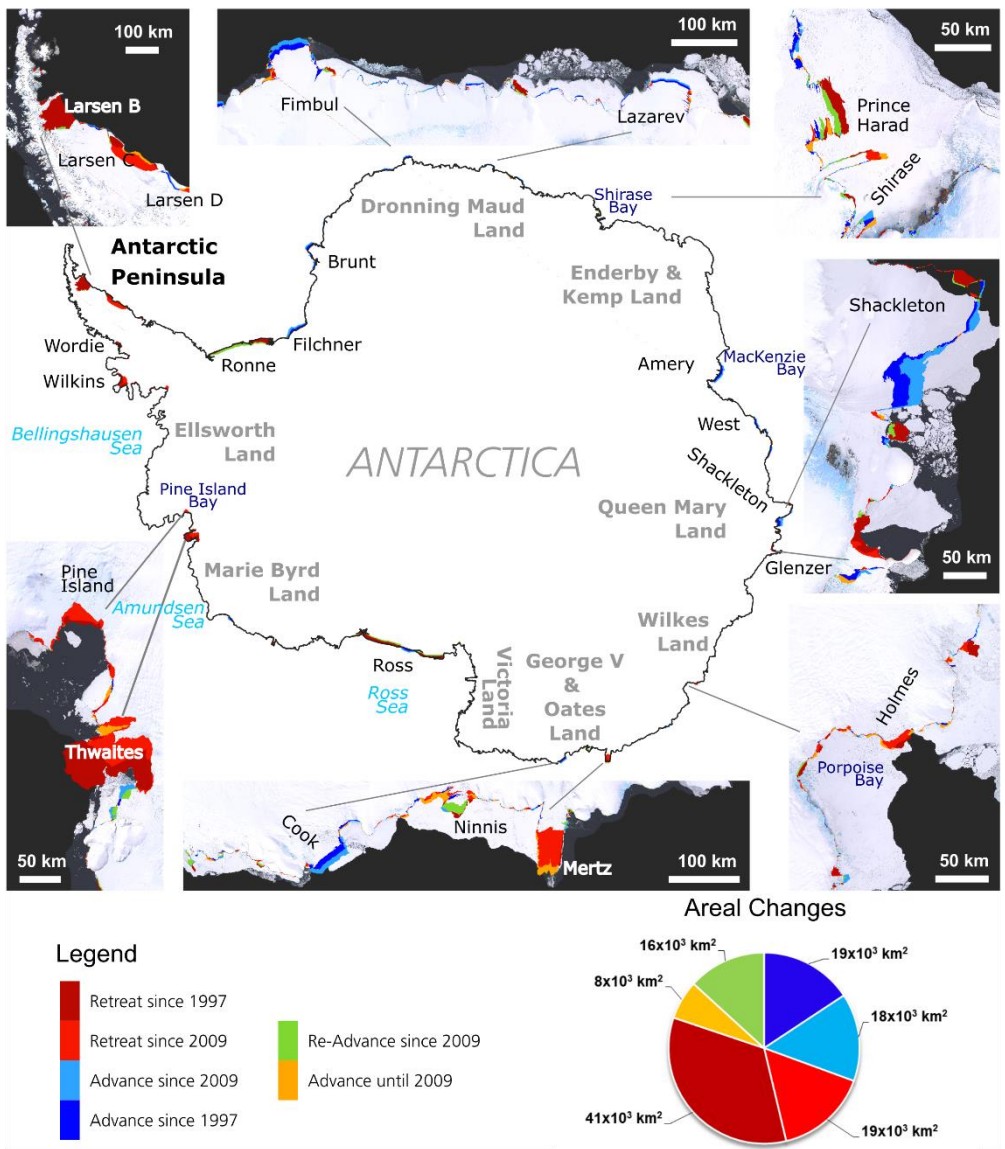

**Figure 1 Coastal change of Antarctica between 1997 and 2018 with enlarged views (counter-clockwise) for Larsen C Ice Shelf, Pine Island Bay, Oates Land, Wilkes Land, Shackleton Ice Shelf, Shirase Bay and Dronning Maud Land. Colours indicate the timing of retreat and advance. The pie chart visualizes losses and gains in the area of the Antarctic Ice Sheet with regard to the indicated year. Labels of coastal sectors are shown in grey, seas in light blue, bays in dark blue and ice shelves and glaciers in black and white fonts. Background: LIMA Landsat Mosaic.**

## 2 Processed and Analysed Data Sets

### 2.1 Coastlines

To calculate the decadal retreat and advance of Antarctic glaciers and ice shelves, we use calving front positions of three Antarctic coastline products from 1997, 2009 and 2018. Timewise, we refer to first (1997-2008) and second (2009-2018) decade even though the time periods are 11 and 9 years, respectively, limited by the availability of coastline products. The first coastline product was automatically extracted from the high-resolution Radarsat-1 mosaic by adaptive thresholding and finalized by manual correction (Liu and Jezek, 2004). The Radarsat-1 mosaic imagery was acquired between September and October 1997 with a spatial resolution of 25 m. The entire dataset is freely available at the National Snow and Ice Data Center (NSIDC) from https://nsidc.org/data/NSIDC-0103/versions/2 (Jezek et al., 2013). Glacier and ice shelf fronts for the year 2009 were manually delineated from the MOA 2009 Surface Morphology Image Map acquired during austral summer 2008/2009 (Nov-Feb) with a spatial resolution of 125 m (Scambos et al., 2007). This dataset is also freely available at NSIDC from https://nsidc.org/data/NSIDC-0593/versions/1 (Haran et al., 2014). The coastline for 2018 was automatically extracted via a fully convolutional network from Sentinel-1 mid resolution (40 m) dual-pol imagery (see methods section). Directly calculating coastal change between these coastline products includes changes in floating calving fronts and grounded ice walls (as shown in Figure 1) even though the amount of change from grounded termini is small and prone to inaccuracies due to the low spatial resolution of the coastline product from 2009.

### 2.2 ERA 5 Reanalysis Data

We use ERA5-Land monthly averaged atmospheric reanalysis data for air temperature and snowmelt information (Copernicus Climate Change Service, 2019b). It is the most state-of-the-art reanalysis product from the European Centre for Medium-Range Weather Forecasts (ECMWF) replacing the former ERA-Interim product (Hersbach et al., 2020). Data is available from 1981/1982 to present day at a 9 km spatial resolution. As ground-based meteorological observations are scarce over the Antarctic continent we decided to use the reanalysis data even though modelled data is less accurate compared to in-situ measurements. Studies comparing in-situ observations to modelled data prove that ERA5 surface air temperature (2 m temperature) outperforms the former ERA-Interim product. ERA5 temperature data has the ability to capture annual variability and the magnitude of temperature change over the Antarctic Ice Sheet (Tetzner et al., 2019; Gossart et al., 2019). The mean absolute error is 2.0° C with higher accuracies in the coastal areas and less accurate results in the interior of the ice sheet (Gossart et al., 2019). Tetzner (2019) report much higher accuracies for the Antarctic Peninsula with a mean absolute error of -0.13° C. Compared to the mean, the variability of temperature is captured with high accuracy at a mean Pearson correlation coefficient of 0.98 (Gossart et al., 2019; Tetzner et al., 2019). We divided temperature measurements into the cooler half ("winter" Apr-Sep) of the year and warmer half ("summer" Oct-Mar) to separate the environmental forcing for different seasons.

Snowmelt data should be handled with care as accuracy assessments for the ERA5 Land snowmelt product were not yet performed. Surface mass balance (SMB) data including modelled snowmelt data was found to slightly underestimate the SMB (Gossart et al., 2019). Snowmelt is calculated within the summer months December, January and February where most melt occurs. The amount of snowmelt is calculated in mm water equivalent (mm w. eq) per day.

The Antarctic continent is circled by weak and irregular easterly winds driven by high pressure areas over the interior of the Antarctic continent created by cold and dry air. Easterlies weaken in the case of a positive Southern Annular Mode (SAM) as the west wind drift shifts poleward. In order to asses those changes in wind direction, we use ERA5 monthly averaged zonal (West to East) wind speed estimations at 10 m above the surface with a lateral resolution of approx. 31 km. ERA5 captures the spatial variability of near-surface wind speed but underestimates strong winds and coastal winds whereas the wind speed

over the interior is captured very accurately during summer (Gossart et al., 2019). Over the ocean, ERA5 annual mean zonal wind speed is considerably underestimated compared to ASCAT (Advanced Scatterometer) observations (Belmonte Rivas and Stoffelen, 2019). Overall, the mean absolute error over the continent is 2.8 m/s with high variance in space and time but an accurate representation of the annual variability in zonal wind speed is achieved (Gossart et al., 2019). ERA5 monthly averaged data on single levels is freely available at the Copernicus Climate Change Service Climate Data Store (CDS) (Copernicus

Climate Change Service, 2019a). Zonal Wind was calculated for the summer months (DJF) because easterly winds show a weakening trend during summer but not throughout the entire year (Hazel and Stewart, 2019).

## 2.3    Sea Ice Days

The most recent Global Sea Ice Concentration Climate Data Record (Version 2) (Lavergne et al., 2019) was downloaded from the Ocean and Sea Ice Satellite Application Facility (OSI SAF, 2017). We used the daily product (OSI-450, OSI-430-b) during

sea ice months April through October in line with previous studies (Massom et al., 2013; Miles et al., 2016). The product covers the time period 1982 to 2018 and is derived from passive microwave data acquired by Nimbus 7 and DMSP satellites. The final sea ice concentration is computed by the passive microwave data in combination with ERA-Interim data. The standard deviation of mismatch between OSI products and ice chart analysis on sea ice concentration is 8 % for ice and open water during winter (JJA). The trend in sea ice extent is very similar between OSI and ice chart products (Brandt-Kreiner et

al., 2019) which allows the assumption of a very accurate data set. The product has a resolution of 25 km x 25 km. In order to calculate actual sea ice days we count each pixel per day with a sea ice concentration higher than 15 % as suggested by previous studies (Miles et al 2016, Massom et al 2013). During the early acquisition years data gaps occur and second-daily acquisitions exist until mid 1987. In this case, we multiplied the monthly available sea ice days by the proportional amount of missing days. We decided not to use data for the year 1986 as data for entire months were missing and mean sea ice days per year could not

be calculated accurately.

## 2.4 Sea Surface Temperature

Sea surface temperature measured by satellite sensors is also provided by the CDS (Copernicus Climate Change Service, 2019a). The most up-to-date product Level 4 (Version 2) consists of multiple satellite observations from NOAA, ERS, Envisat and Sentinel-3 satellites with a 0.05 ° gridded resolution (approx. 5.5 km). We calculated surface temperatures only for the months with little to no sea ice cover (Oct-Mar). This reduces errors over sea ice where the thickness and concertation of sea ice can account for large measurement errors (Kwok and Comiso, 2002). Uncertainties of Level 4 data vary depending on the year of acquisition and latitude of measurement. The median difference compared to in-situ drifter measurements is up to -0.4°Cfor low latitudes during the early acquisition years before 1996. Afterwards the accuracy increases to better than -0.1°C for low latitudes. A stability assessment of the Level 4 product calculated a maximum trend of 0.01°C per year (Embury, 2019). A summary of all processed climate variables is given in Table 1.

**Table 1 Summary of processed climate variable data sets.**

| Climate Variable | Time Span | Season | Spatial Resolution | Accuracy | Data | Data Store |
|---|---|---|---|---|---|---|
| **Air Temperature** | 1982-2018 | Summer (Oct-Mar) Winter (Apr-Sep) | 9 km | 0.13 – 2.0 °C | modelled | CDS |
| **Snowmelt** | 1982-2018 | Dec-Feb | 9 km | - | modelled | CDS |
| **Zonal Wind** | 1982-2018 | Dec-Feb | 31 km | 2.8 m/s | modelled | CDS |
| **Sea Ice Days** | 1982-2018 (not 1986) | Apr-Oct | 25 km | 8 % std to ice chart | modelled + satellite | Osisaf |
| **Sea Surface Temperature** | 1982-2018 | Oct-Mar | ~5.5 km | -0.1 to -0.4°C | multiple satellites | CDS |

## 3 Methods

### 3.1 Extraction of Sentinel-1 Coastline

We use a modified version of the automatic coastline extraction approach published by Baumhoer et al. (2019) to extract the Antarctic coastline for 2018. To cover the entire Antarctic coastline, 158 dual-pol and 17 single-pol medium resolution Sentinel-1 scenes were processed. Dual polarized scenes contain radar backscatter values for the polarizations HH and HV whereas single polarized scenes only include the HH polarization. The coastline was split into 18 zones based on ice flow divides, defining major ice sheet basins after Rignot et al. (2011). For each zone all available scenes acquired during winter months (Jun-Aug) in 2018 were selected. Depending on the scene availability, each zone was covered at least by one Sentinel-1 scene and in the best case by three scenes. In case no dual-pol scenes were available single-pol data was selected. First, each scene was pre-processed (thermal correction, calibration, terrain correction), masked with a coastline buffer of 100 km and tiled into 780x780 pixel tiles. The convolutional neural network (CNN) U-Net was used to segment each tile into the class ocean and the class land ice. We trained the U-Net on 15 different Antarctic coastal regions during various seasons with 40,036 image tiles from 75 Sentinel-1 scenes. For single-pol scenes we re-trained our network for HH polarized images only. This

decreased the accuracy slightly as only one polarization limits the information on surface backscatter characteristics. In the post-processing step the mean of all segmented tiles within one zone was calculated and then thresholded by a prediction probability of 50 % for the class land ice. In case of multi-coverage by several satellite scenes more robust results were obtained by merging the prediction probabilities. Morphological filtering and the exclusion of higher ice sheet areas by integrating elevation information from the TanDEM-X Polar DEM 90 (Wessel et al., 2021)reduced further errors. Finally, from the binary segmentation results the border between both classes was extracted as the final coastline.

## 3.2    Derivation of Calving Front Change

Calving front change was estimated by calculating the change in area between the coastlines in 1997, 2009 and 2018 over the area of floating glacier tongues and ice shelves excluding grounded glacier termini. As coastline extraction and delineation is a very subjective task, and all coastlines originated from different sources and resolutions, deviations occurred in many areas. Areas of fast ice, mélange and icebergs trapped in sea ice were error sources. For the automatically extracted coastline 2018, errors existed along the western Antarctic Peninsula and in areas where only single-pol imagery was available. The MODIS derived coastline often included snow covered sea ice which is difficult to distinguish from glacier ice in optical imagery. To minimize errors and mismatches all three coastlines were manually corrected and adjusted. Each coastline product was corrected based on the satellite imagery from which they were originally created. After manual correction any change in area between the coastline products could be attributed to glacier front change. From each coastline a raster (resolution 40 x 40 m) was created with a unique value over the ice covered area. All three raster layers were stacked and summed up, so each raster value pixel is associated with retreat or advance of the specific year. The area change was calculated for each major floating ice shelf and glacier tongue wider than 3 km based on a 40 m resolution raster in Polar Stereographic Projection. To account for inaccuracies in the manual adjustment of all coastlines, we measured the accuracy as the area change over 30 randomly picked stable coastline areas. Any change over those regions can be attributed to errors in manual delineation, imagery resolution differences and errors in orthorectification of the different satellite image mosaics. The error is calculated per km of the coastline and calculated in proportion to the measured front length. On average, the coastlines deviated ±1.2 pixels per kilometre per year. Broken down for each coastline product, the error between 1997-2009 was 0.4 pixels per km per year, 2.3 pixels between 2009-2018 per km per year and 0.9 pixels between 1997 -2018 per km per year.

## 3.3    Climate Data Correlation

In order to assess the influence of climate variables (such as air temperature, sea ice days, sea surface temperature, snowmelt and zonal wind) on glacier and ice shelf retreat, we spatially correlated the percentage of advance and retreat for each minor glacier basin (as defined by Mouginot (2017)) with the decadal mean value of each climate variable. Means were calculated from 37-years of climate data for the three time periods 1982 – 1996, 1997 – 2008, and 2009-2018. The first period is used as the reference period to measure temporal changes in the following two decades. Hence, the inputs of the correlation covered the two time spans 1997-2008 and 2009-2018 for which also the glacier retreat was calculated. To also assess relative changes

in the variables to previous times we subtracted the reference mean (1982-1996). This means the relative values indicate the change of an environmental variable within the first/second decade compared to the reference time frame. Zonal wind, sea ice coverage and sea surface temperature means were calculated within a 100 km seawards buffer along the coastline. Mean air temperature and snowmelt was calculated within a 100 km buffer landwards from the coastline covering the surface of ice shelves. The 100 km landwards buffer was chosen as the best trade-off to calculate averages over the ice shelf area by not including too much of the ice sheet. Due to a circum-Antarctic analysis this is only a rough approximation and especially over smaller glaciers within steep terrain the average not only includes changes over the glacier tongue but the surrounding area. Again, relative values were calculated by subtracting the reference mean 1982-1996. To remove the effect of different basin sizes and different amounts of ice discharge we took the percentage of advance and retreat within each basin instead of the absolute value for the correlation. Input data for the Pearson correlation were the mean of all assessed climate variables (absolute and relative averages) as well as the percentage of retreat and advance. This created 14 different variables which were correlated with each other based on 188 observations (N=188). The number of observations is derived from 94 assessed glacier basins with variable averages for the two different decades (1997-2008 and 2009-2018).

## 4    Results

### 4.1    Advance and Retreat of Antarctic Glaciers and Ice Shelves

Circum-Antarctic glacier and ice shelf front changes were assessed by comparing Antarctic coastline products during 1997-2008 and 2009-2018. The results are shown in Figure 2. Between 1997 and 2008 ice shelf and glacier extents decreased by -29,618±1,193 km$^2$ where 69 % of the total changed area retreated and 31 % advanced (see Table 2 for all change rates). In contrast, during the period 2009 to 2018 a slight area increase of 7,108±1,029 km$^2$ could be observed with 44 % of the total changed area retreating and 56 % advancing. The locations of area change are almost similar for both observation periods. Ice shelves along the Antarctic Peninsula retreated over the entire observation period (1997-2018). The only exception was the advance of the Larsen D Ice Shelf which is located close to the Ronne Ice Shelf in the Weddell Sea (see Figure 2). Between 1997 and 2008, the large disintegration events of the Larsen B, Wilkins and Wordie ice shelves resulted in a 37 % higher calving amount from the Antarctic Peninsula, as compared to the amount calved from this region between 2009-2018. During the second decade the breakup of iceberg A-68 from the Larsen C Ice Shelf and the further disintegration of the Wilkins Ice Shelf were regions of strong area loss. In total, the rate of retreat along the Antarctic Peninsula was 5-6 times higher than glacier advance in this region over the last 20 years.

West Antarctica lost more than three-times as much calving area in the first decade (1997-2008) compared to 2009-2018. In West Antarctica, 75 % of glacier and ice shelf retreat can be attributed to the calving of Ronne and Ross West ice shelves during the first decade. The retreat at WAIS was significantly lower in the second decade whereas the Ross and Ronne ice shelves started to advance again. Excluding these two ice shelves, the rest of West Antarctica experienced predominant glacier and ice shelf retreat during both observation periods as shown in Figure 2. Pine Island and Thwaites glaciers retreated over the

250 entire observation period (1997-2018). The Getz and Abbot ice shelves had stable front positions in the first decade but started to retreat after 2008. Only the Crosson Ice Shelf showed a contradicting pattern with retreat in the first and re-advance in the second decade.

**Table 2 Retreat and advance rates of Antarctic glaciers and ice shelves for each basin between 1997 and 2018 (upper table). Total**
255 **lost and gained ice shelf/glacier area per decade (lower table). For basin abbreviations see Figure 2.**

| | basin | 1997-2008 | | 2009-2018 | | 1997-2018 | |
|---|---|---|---|---|---|---|---|
| | | advance ($km^2$/yr) | retreat ($km^2$/yr) | advance ($km^2$/yr) | retreat ($km^2$/yr) | advance ($km^2$/yr) | retreat ($km^2$/yr) |
| EAIS | A-Ap | 249±14.9 | 164±17.4 | 295±27.5 | 112±13 | 215±11.8 | 85±6.5 |
| | Ap-B | 62±4.8 | 89±5.6 | 80±8.9 | 53±4.2 | 34±3.8 | 36±2.1 |
| | B-C | 187±1.9 | 4±2.2 | 186±3.5 | 20±1.6 | 179±1.5 | 4±0.8 |
| | C-Cp | 380±7.7 | 167±9 | 363±14.2 | 153±6.7 | 323±6.1 | 111±3.3 |
| | Cp-D | 49±4.6 | 58±5.4 | 45±8.6 | 97±4.1 | 23±3.7 | 51±2 |
| | D-Dp | 154±5 | 64±5.8 | 156±9.2 | 317±4.3 | 85±3.9 | 111±2.2 |
| | Dp-E | 68±9.5 | 58±11.1 | 74±17.5 | 81±8.3 | 34±7.5 | 32±4.1 |
| | E-Ep | 6±3 | 411±3.5 | 156±5.5 | 8±2.6 | 5±2.4 | 154±1.3 |
| | Jpp-K | 163±2.1 | 2±2.4 | 252±3.8 | 2±1.8 | 203±1.6 | 1±0.9 |
| | K-A | 306±7.8 | 23±9.1 | 285±14.3 | 82±6.8 | 270±6.2 | 24±3.4 |
| | **EAIS** | **1626±61.1** | **1040±71.4** | **1894±112.9** | **926±53.4** | **1370±48.6** | **607±26.6** |
| AP | Hp-I | 20±4.5 | 328±5.3 | 25±8.4 | 305±4 | 11±3.6 | 306±2 |
| | I-Ipp | 87±3.9 | 938±4.6 | 59±7.2 | 592±3.4 | 12±3.1 | 715±1.7 |
| | Ipp-J | 91±4.5 | 67±5.3 | 118±8.4 | 75±4 | 71±3.6 | 38±2 |
| | **AP** | **198±13** | **1333±15.2** | **202±24** | **972±11.4** | **94±10.3** | **1060±5.7** |
| WAIS | Ep-F | 160±7.7 | 868±8.9 | 452±14.1 | 74±6.7 | 136±6.1 | 341±3.3 |
| | F-G | 103±5.2 | 101±6.1 | 73±9.7 | 106±4.6 | 59±4.2 | 73±2.3 |
| | G-H | 40±3.6 | 371±4.2 | 49±6.7 | 369±3.2 | 14±2.9 | 340±1.6 |
| | H-Hp | 26±3.7 | 35±4.3 | 6±6.8 | 97±3.2 | 3±2.9 | 51±1.6 |
| | J-Jpp | 29±5.8 | 1127±6.8 | 643±10.7 | 26±5.1 | 13±4.6 | 317±2.5 |
| | **WAIS** | **358±26** | **2502±30.4** | **1222±48** | **672±22.7** | **225±20.7** | **1121±11.3** |

| | 1997-2008 | | 2009-2018 | | 1997-2018 | |
|---|---|---|---|---|---|---|
| region | advance ($km^2$) | retreat ($km^2$) | advance ($km^2$) | retreat ($km^2$) | advance ($km^2$) | retreat ($km^2$) |
| WAIS | 3942±286 | 27525±334 | 11612±456 | 6382±216 | 4617±424 | 22970±232 |
| AP | 2178±143 | 14660±167 | 1920±228 | 9232±36 | 1929±212 | 21722±116 |
| EAIS | 17886±672 | 11439±785 | 17990±1072 | 8801±51 | 28089±997 | 12453±545 |
| AIS | 24006±1100 | 53624±1286 | 31522±1756 | 24414±303 | 34636±1632 | 57146±893 |
| **difference** | **-29618±1193** | | **7108±1029** | | **-22510±1263** | |

The glaciers and ice shelves of the EAIS showed an overall stable advancing tendency with similar rates for both decades with 2638±418 $km^2$ more retreat during the first (attributed to calving of Ross East Ice Shelf) and no clear change in advance (104±872 $km^2$) during the second decade. Strongest advance was observed for the Amery and Filchner ice shelves as well as
at the shelves of Dronning Maud Land and Queen Mary Land (see Figure 1 for location). During 1997-2008, clear retreat appeared for the Ross East Ice Shelf and along Enderby Land especially in Shirase Bay. Wilkes Land as well as Victoria Land shared equal proportional amounts of retreat and advance. Between 2009 and 2018, the Ross Ice Shelf and Enderby Land entered a phase of advance whereas Wilkes Land, George V and Adélie Land started to retreat predominately.

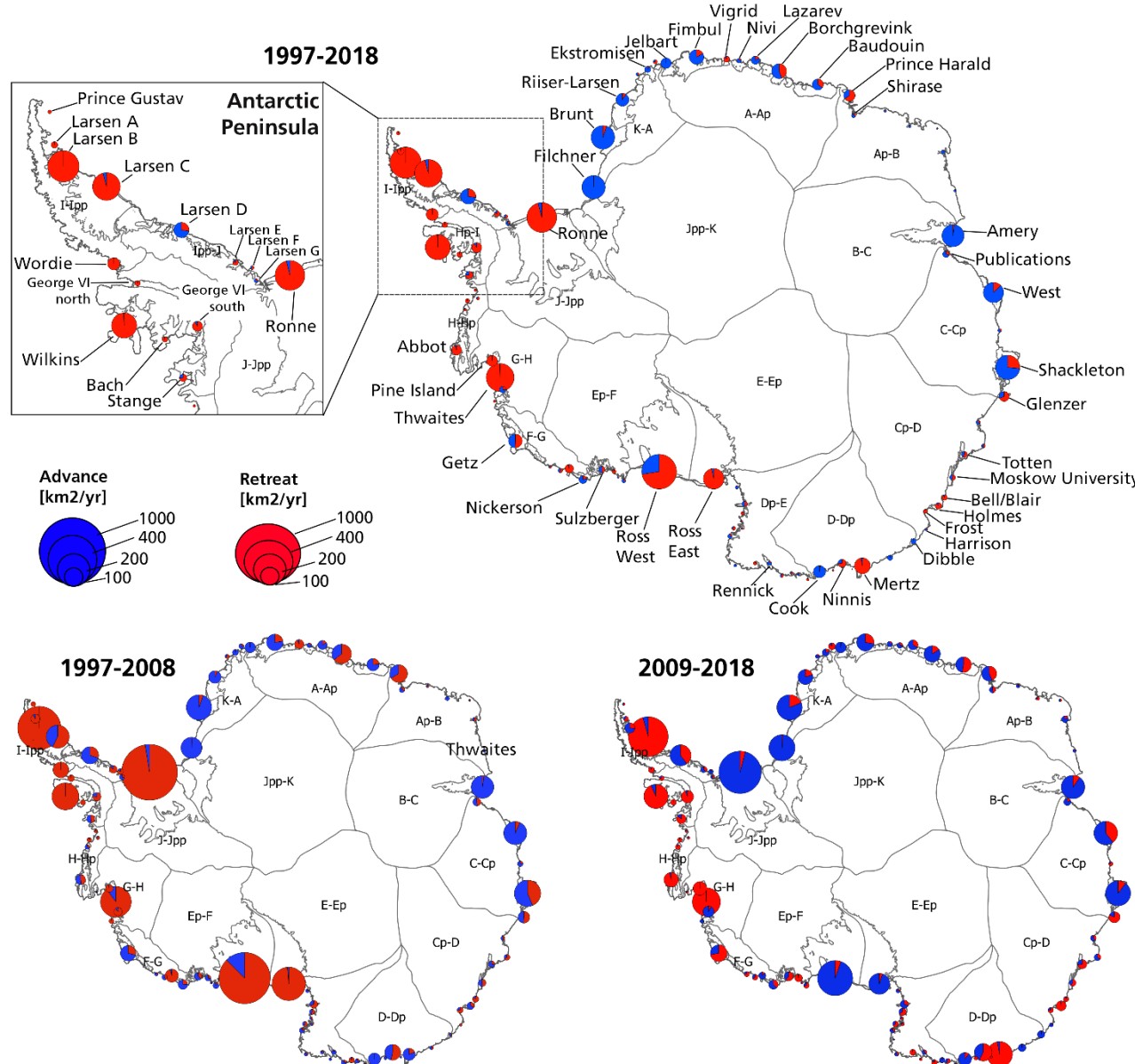

**Figure 2 Glacier and ice shelf extent changes for major glaciers and ice shelves over the last two decades. Circles indicate the rate of retreat or advance. Major ice sheet basins as defined by Rignot et al. (2011).**

## 4.2 Air Temperature

The mean Antarctic air temperature was equal for both decades during winter but in summer +0.6 °C higher in the second decade (2009-2018). However, the overall mean does not reflect the strong regional differences in air temperature changes as

shown in Figure 3. During summer and winter the Antarctic Peninsula and the adjoining coast along Bellingshausen and Amundsen Sea Sector as well as Queen Mary Land (EAIS) were cooler between 2009-2018, whereas the interior of the ice sheet warmed during this period. Compared to the reference mean (1982-1996) summer temperatures between 1997-2008 were cooler except for the Peninsula, Queen Mary Land, and the Ross West and Filchner ice shelves. In the second decade this changed in sign where the Antarctic Peninsula started to cool down and the interior increased in temperature. For the winter months almost over the entire Antarctica continent a warming tendency was observed compared to the reference mean. Only exceptions were Victoria, Oates and George V Land (all EAIS). Additionally, over the Getz Ice Shelf (WAIS) cooler temperatures by 1°C occurred in 2009-2018 compared to the reference mean. For Dronning Maud Land (EAIS) a strong change was observed as the ice shelf surfaces warmed up to 1.8°C in the second decade.

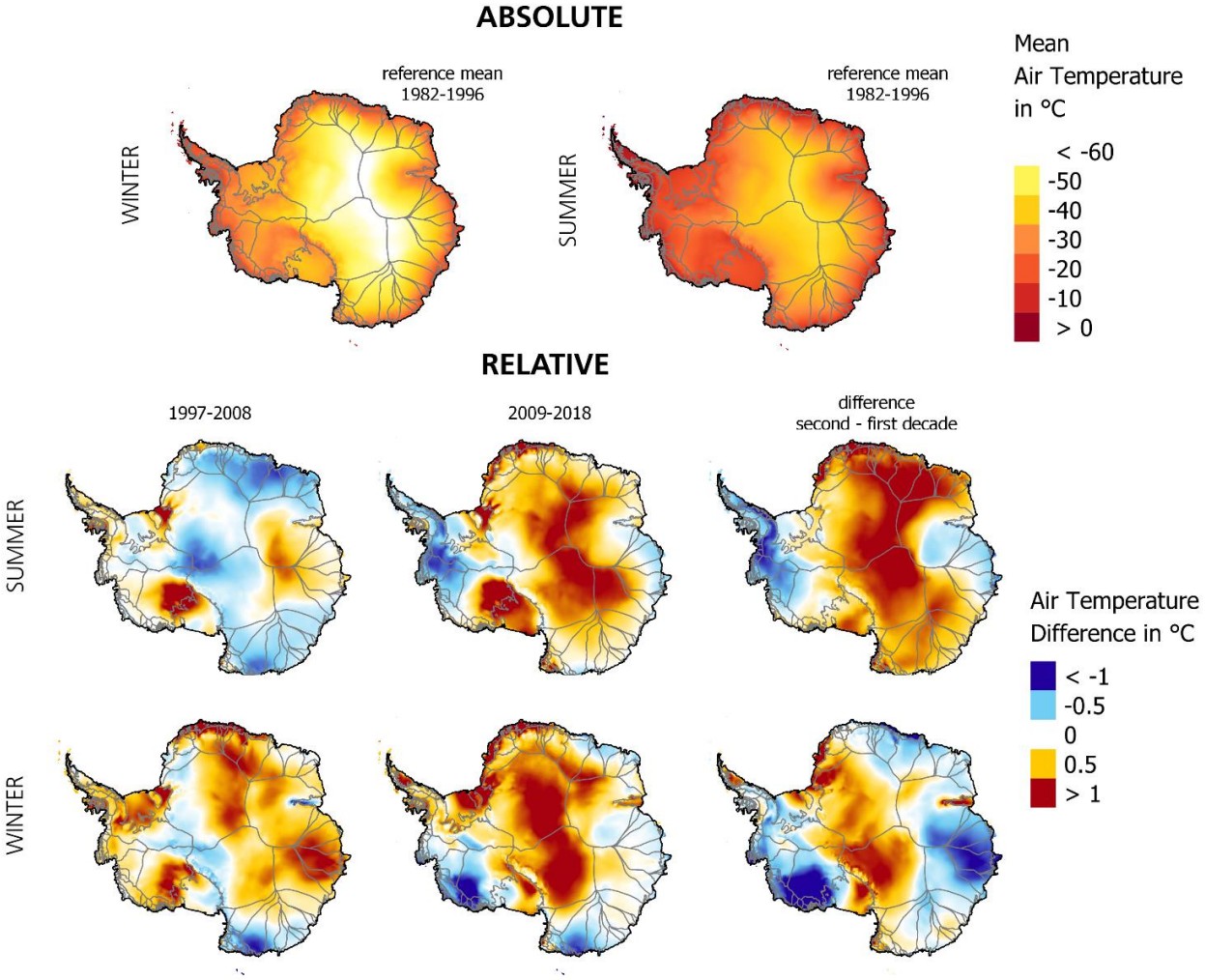

**Figure 3 Absolute air temperature for the reference mean measured between 1982 and1996 and relative changes compared to the reference mean. Additionally, the difference between the first and second decade is illustrated.**

### 4.3 Sea Ice Days

Changes in sea ice days are shown in Figure 4 and absolute numbers for sea ice days are provided in the supplement in Figure S1. A decrease or increase in sea ice days refers to the difference in the number of sea ice days per year (during the sea ice season Apr-Oct) compared to the reference mean. The strongest decrease in sea ice days could be observed at the northern tip of the Antarctic Peninsula with a shorter sea ice cover per year of up to 40 days during the first decade compared to the reference mean. During 1997 and 2008 a decrease in sea ice days was observed along the Bellingshausen Sea Sector with up to 20 days less and in the Amundsen Sea Sector with up to 5-10 days less. Compared to the first decade, the mean sea ice cover persisted longer in the second decade along the Antarctic Peninsula and the Bellingshausen Sea Sector whereas in the Amundsen Sea Sector the sea ice days decreased up to 10 days further compared to the first decade. Along East Antarctica, sea ice coverage increased 10 days on average except for Dronning Maud Land with a slight decrease of up to 5 days. When comparing the two decades more sea ice days occurred during 2009 to 2018 along East Antarctica and the northernmost Antarctic Peninsula whereas the decrease along West Antarctica continues. Changes in sea ice days were less extreme along East Antarctica with ±10 days in difference compared to 1982-1996. Between 1997-2008 the duration of sea ice cover was up to 5 days shorter along Dronning Maud and Wilkes Land compared to the reference mean. In the second decade sea ice days along entire East Antarctica increased with a highest increase along Wilkes Land with up to 10 days (compared to 1982-1996).

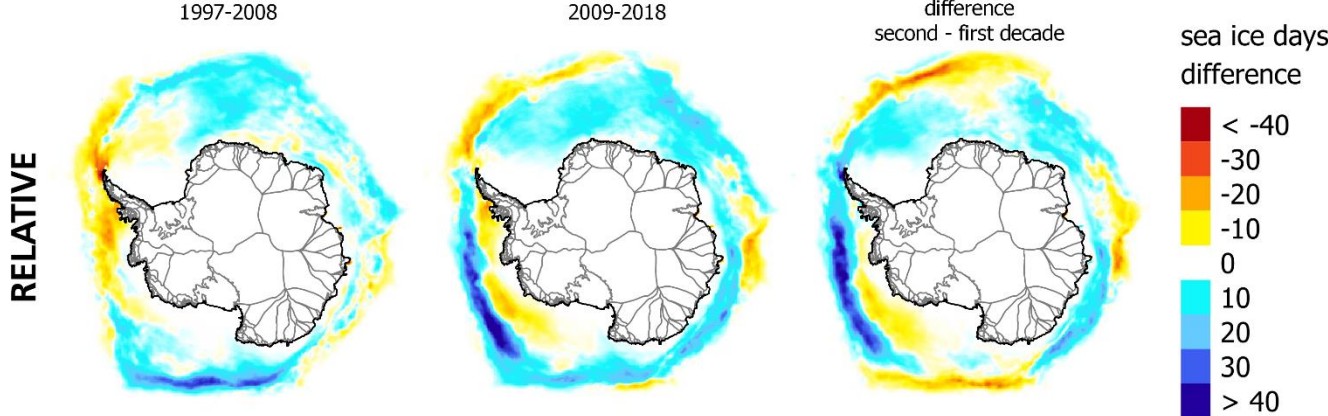

**Figure 4 Difference in the number of sea ice days per year (during the sea ice season Apr-Oct) compared to the reference mean (1982-1996) and the difference in sea ice cover duration between both decades.**

### 4.4 Sea Surface Temperature

The changes in sea surface temperature during months with little sea ice cover (Oct-Mar) relative to the reference mean are shown in Figure 5 (absolute values in Figure S2). Sea surface temperatures of the Southern Ocean cooled (~ -0.5°C) along East Antarctica and warmed along West Antarctica and the western Antarctic Peninsula (~ +0.5°C) within the last two decades compared to 1982-1996 which is above the data uncertainty of 0.1 to 0.4°C. The warming was less strong in the first decade and intensified within the second decade, especially in the Bellingshausen Sea Sector, with maxima at George VI north

(+0.65°C) and Pine Island Bay (+0.35°C), and with a slightly weaker increase along the Amundsen Sea Sector (+0.15°C). The cooling along the East Antarctic was less strong in the second decade. Along East Antarctica for Amery Ice Shelf (+0.35°C), West Ice Shelf (+0.2°C) and Shackleton Ice Shelf (+0.25°C) warming was observed compared to the first decade.

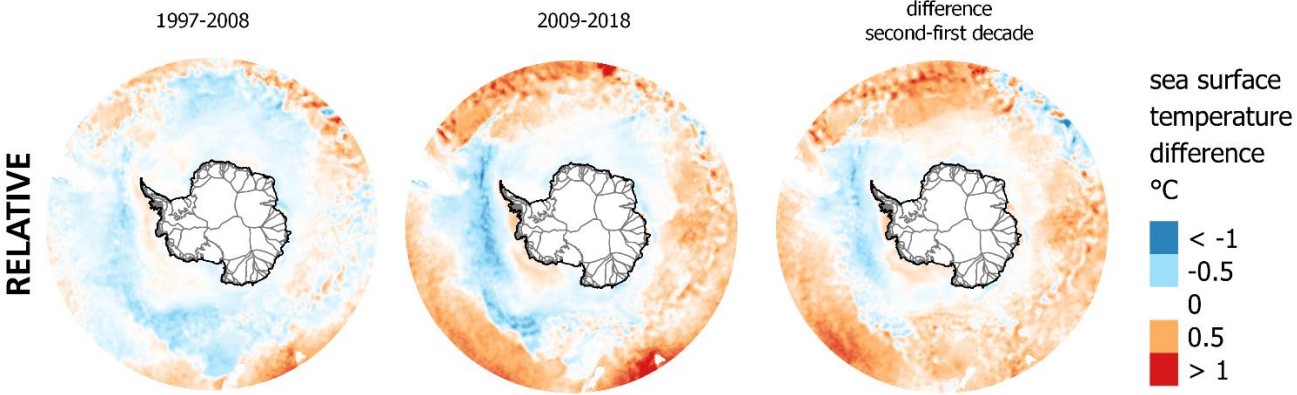

**Figure 5 Mean sea surface temperature changes (Oct-Mar) compared to 1982-1996 and the difference in sea surface temperatures between both decades.**

### 4.5    Snowmelt

Snowmelt over the Antarctic Peninsula (see Figure 6) was more extensive during the reference mean compared to the recent
two decades. This is the reason why relative snowmelt in 1997-2008 and 2009-2018 was mostly negative, with up to -0.3 mm w. eq. per day. Nevertheless, strong snowmelt also occurred during the recent two decades, but predominantly over the Wilkins and Larsen B ice shelves, as well as at the northern tip of the Antarctic Peninsula. During the reference mean, snowmelt was greatest on the Antarctic Peninsula, Ronne, Abbot, Shackleton and Amery ice shelves as well as Shirase Bay. In the first decade snowmelt expanded to Pine Island Bay, the Getz and Ross ice shelves as well as along Wilkes, George V and Dronning Maud
Land in East Antarctica. In all cases, the increase in melt was small with 0.1 mm w. eq. per day.  Within the most recent decade melt became more extensive (+0.1 mm w. eq. per day) over George V, Oates and parts of Wilkes Land as well as over Getz and Sulzberger ice shelves. Smaller areas of strong surface melt in Pine Island Bay and Dronning Maud Land were only observed during the first decade.

### 4.6    Zonal Wind

Figure 7 shows changes in zonal wind around the Antarctic continent (absolute values in Figure S3). In the first decade, weakening easterlies (~ +0.5 m/s) were observed along the Antarctic Peninsula, Bellingshausen and Amundsen Sea Sector, as well as in East Antarctica in the area of Shackleton Ice Shelf. Stronger easterlies occurred only at George V and Dronning Maud Land (both EAIS). Within the second decade the strengthening westerlies along East Antarctica expanded from Amery Ice Shelf to Victoria Land with up to + 1m/s. Additionally, the westerlies strengthened in the Bellingshausen Sea Sector but

weakened in the Amundsen Sea Sector. Strong dominating easterlies occurred along Dronning Maud Land with up to -0.75 m/s within the second decade.

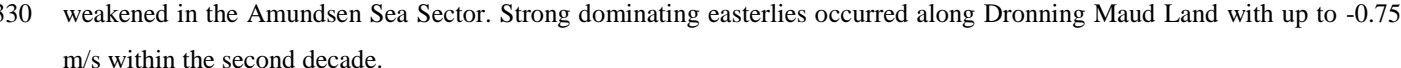

**Figure 6 Mean snowmelt over Antarctica with enlarged views of the Antarctic Peninsula. Snowmelt is given in mm water equivalent per day. The mean snowmelt was calculated over 90 days per year during the summer months December to February**

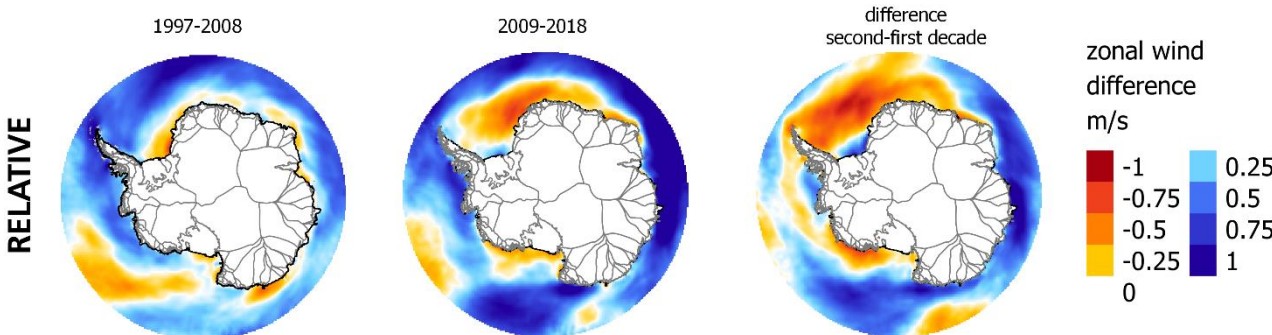

**Figure 7 Zonal Wind (West to East) around the Antarctic continent in m/s for 1997-2008 and 2009-2018 compared to the reference mean. Additionally, the difference in wind speed between both decades is illustrated. Positive shifts in zonal wind indicate stronger westerlies with the potential to cause upwelling of warm Circum Polar Deep Water.**

## 4.7    Southern Annular Mode (SAM)

Changes in zonal wind direction are closely connected to fluctuations in SAM. Positive phases of SAM weaken the easterlies around the Antarctic continent and shift the westerlies poleward closer to the coastline. Phases of a positive SAM occur when air pressure over the Antarctic Ice Sheet lowers but rises over the subtropical ocean. Figure 8 shows the evolution of the SAM Index since 1960 based on calculations from Marshall (2003). The 5-year moving average (in red) of the SAM has been positive since 1992, with positive values during the first and second decade. Annual peak events with exceptional high SAM occurred in 1997/1998 and 2014/2015 shown in blue (annual) and orange (summer only). Note the shift in peaks as the summer value for SAM averages values from December to February (December indicates the year).

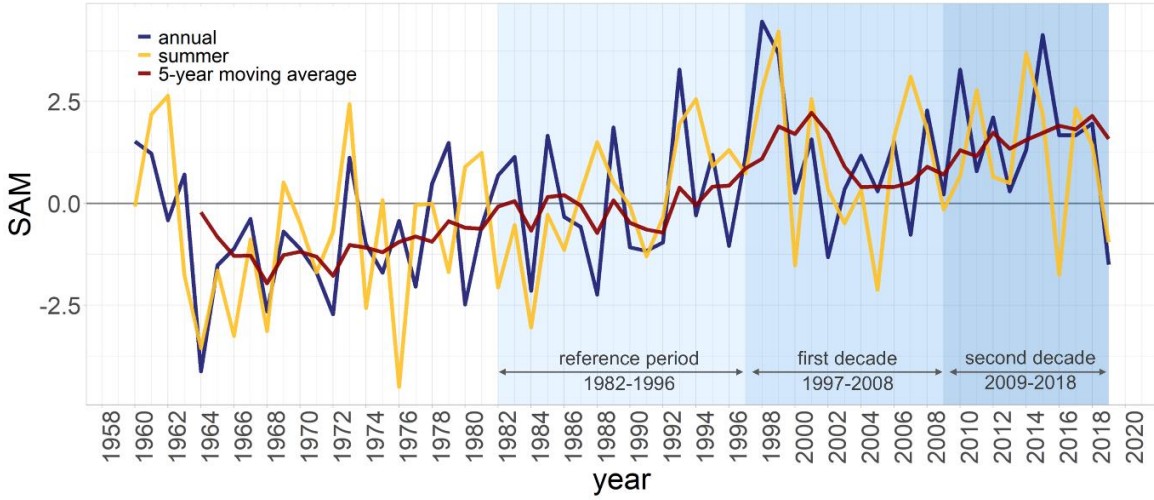

**Figure 8 Southern Annular Mode (SAM) since 1960. Values are given for the annual SAM (blue), SAM during summer (orange) and a 5-year moving average for the annual SAM (red). The blue background indicates the investigated periods as delimited by the available coastline and climate data. For SAM during summer (Dec-Feb), the beginning of austral summer (hence December) indicates the year of the peak. Data: Marshall et al, (2018).**

## 4.8    Correlation between Climate Variables and Calving Front Position

Potential drivers of calving front retreat were diagnosed by correlating the analysed climate variables with the percentage of retreat/advance within each glacier/ice shelf basin. This allows for the assessment of a potential spatial relationship between calving front retreat and changes in environmental variables. The results of the Pearson correlation are displayed in Figure 9. Dark blue colors indicate a strong positive, dark red color a strong negative correlation. Stars indicate a significant Pearson correlation with stars for p=0.05 (*), 0.01 (**) and 0.001(***). Correlations for retreat and advance are counterparts, hence correlations are of the same magnitude for each climate variable but with reversed sign. Overall, weak to moderate correlations with significance occur for relative summer sea surface temperature, absolute air temperature, snowmelt, relative zonal wind speed and sea ice days. The strongest positive linear relationship (r=0.44) exists between calving front retreat and relative sea surface temperature. Slightly weaker is the positive correlation between glacier and ice shelf front retreat and absolute air temperature on the ice shelf surface (r=0.18 for summer, r=0.23 for winter). A relatively more positive zonal wind (hence strengthening westerlies) correlates positively with calving front retreat (r=0.30) but the absolute strength of zonal winds does not. Decreases in sea ice days correlate positively with calving front retreat (r=0.33 (absolute), r=0.27 (relative)). The mean daily amount of snowmelt correlates weakly but significant with glacier and ice shelf front retreat (r=0.17). The correlation of the climate variables among each other reflects that they are closely linked to each other. Higher air (r=-0.26 for summer, r=-0.36 for winter) and sea surface temperatures (r=0.44) have a negative relationship to an increase in sea ice days. An increase in sea ice days negatively correlated with an increase in zonal winds (r=0.31). Stronger snowmelt correlates positively with warmer summer (r=0.46) and winter (r=0.37) air temperatures. An increase of zonal winds was positively related to decreasing summer air temperatures (r=-0.31 for summer).

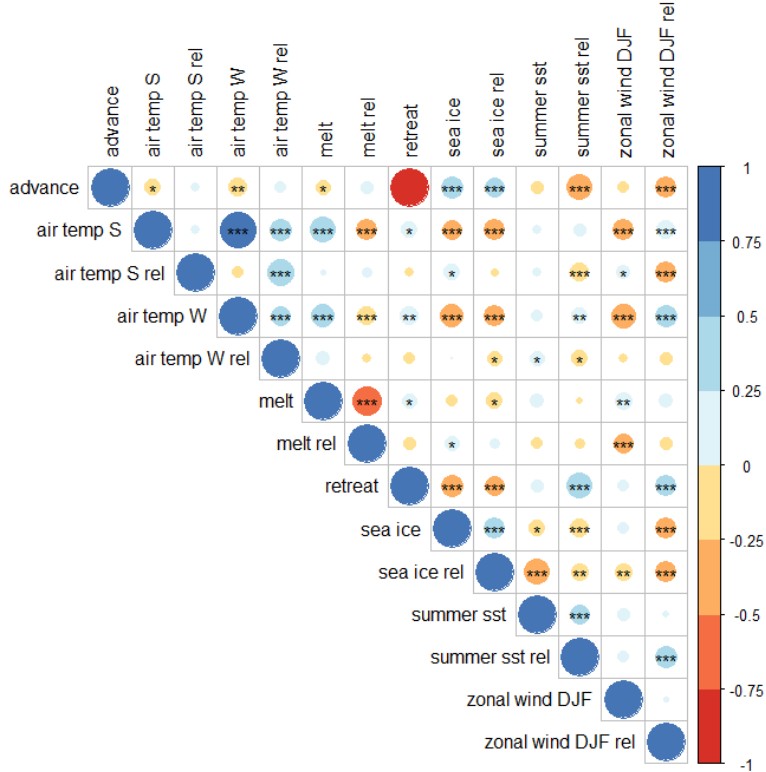

**Figure 9 Correlation between glacier and ice shelf change and the analysed climate variables winter/summer air temperature, snowmelt, sea ice days, summer sea surface temperatures and zonal wind speed. Colour and circle size indicate the correlation coefficient. Stars indicate significance levels for p=0.05 (*), 0.01 (**) and 0.001(***). S: summer, W: winter, rel: relative to 1982-1996, DJF: Dec-Feb.**

## 5    Discussion

For the first time, this study presents circum-Antarctic glacier and ice shelf front change over two decades. The correlation analysis indicates a spatial relationship exists between calving front retreat and strengthening westerlies, higher air temperatures, intense snowmelt, decreasing sea ice cover, and rising sea surface temperature. We want to discuss whether these external environmental factors were responsible for the observed glacier retreat or if internal glaciological forces were the key driver instead. To address this question, two major challenges arise. First, a significant correlation between calving front retreat/advance and climate variables cannot alone provide conclusive evidence of a causal link. Second, the retreat forced by environmental drivers has to be disentangled from retreat caused by internal glaciological and external mechanical factors. To find solid evidence for drivers of glacier and ice shelf retreat, we will discuss observed glacier/ice shelf retreat on the basis of findings from previous studies in combination with measured changes in analyzed climate variables.

## 5.1    Antarctic Peninsula

Along the Antarctic Peninsula a decrease in ice shelf extent resulted mainly by the disintegration and retreat of Larsen B, Larsen C, Wilkins and Wordie ice shelves (see Figure 2). The Wordie Ice Shelf disintegrated in a series of calving events beginning in the 1960s which were controlled by several pinning points (Friedl et al., 2018). The initial destabilization of the ice shelf was likely caused by a shift in flow which caused rift development at the ice rises and a collapse in 1989 (Walker and Gardner, 2017). Additional destabilization occurred through basal melt induced thinning (Depoorter et al., 2013; Friedl et al.,

2018; Rignot et al., 2013; Walker and Gardner, 2017). A later calving event accompanied by thinning and acceleration was associated with upwelling of warm CDW which was strong in the years 2008 to 2011 due to consecutive years of positive SAM (Walker and Gardner, 2017) with a peak in 2010 (SAM Index above 2.5, see Figure 8). In addition to the continuous thinning by basal melt, the retrograde bed in a subglacial deep trough destabilized the Wordie Ice Shelf and led to grounding line retreat in 2010/2011 (Friedl et al., 2018). For the major calving event of the ice shelf in 1989, we observed stronger zonal

winds between 1997-2008 compared to the reference period (+0.62 m/s) and the most recent decade (+ 0.28 m/s) (2009-2018). This means that during the break up event the easterly wind direction weakened towards a more westerly direction. This change in wind-direction causes upwelling of warm CDW and enhances basal melt which is in line with the basal melt observed by the studies above. Additionally, we not only observed a positive SAM during 2008 to 2011, as in Walker and Gardner (2017), but also for the loss of the remaining ice tongues in summer 1998/1999. This strengthens the hypothesis that the retreat of

Wordie Ice Shelf was influenced by environmental forcing through enhanced basal melt. We also observed a decreasing sea ice coverage (-14 day compared to the reference mean) in the area of the former Wordie Ice Shelf which has not been discussed in literature but might have further facilitated the destabilization of the ice shelf margins. Despite strong evidence by observed environmental forcing it shall not be neglected that the glaciers flowing into the former Wordie Ice Shelf are still, at least partly, responding to the collapse initiated in the 1960s.

The Wilkins Ice Shelf also experienced a series of calving events likely caused by environmental and buoyancy forces (Braun and Humbert, 2009; Scambos et al., 2000). Scambos et al. (2000, 2009) linked the calving event in summer 1998 to hydrofracture, that was due to surface melt and ponding caused by atmospheric forcing. In contrast, Braun et al. (2009) speculate that basal melt thinned the boundaries and bending stresses from buoyancy forces caused the break-up in 1998. Additionally, the minimal extent of sea ice was discussed as a potential factor for retreat (Lucchitta and Rosanova, 1998). Our

observations confirm the decrease in sea ice days (-9 days), potential basal melt through enhanced zonal winds (+0.47 m/s), a peak in SAM, increased surface melt (+0.06 mm w. eq. per day), higher sea surface temperatures (+0.13°C) and increased air temperature (+0.58°C) within the first decade compared to the reference time period. This suggests that the calving event in 1998 was probably forced by a combination of several environmental drivers that destabilized and thinned the ice shelf. Whether these factors alone would have caused the retreat or the additional stresses through buoyancy as stated by Braun et

al. (2009) were necessary to initiated the final calving event stay unresolved. A second break up event of the Wilkins Ice Shelf occurred in three stages during February, May and June 2008 (Scambos et al., 2009), and the remaining ice bridge collapsed

in April 2009, likely due to mechanical external forcing by strong winds (Humbert et al., 2010). For the calving events in 2008, Scambos et al. (2009) found surface melt water as the main pre-condition for calving whereas Braun et al. (2009) excluded melt pond drainage as a driver and attribute the calving to fracture development and rift formation due to buoyancy forces and bending stresses through variable ice thickness. As described above the surface melt did only slightly increase between 1997-2008 compared to the reference period which strengthens the hypothesis by Braun et al. (2009). Therefore, we suggest that a combination of glaciological, environmental forced and mechanically forced processes caused the disintegration of Wilkins Ice Shelf.

The disintegration of Larsen B in 2002 is associated with increased surface melt causing enhanced fracturing (Khazendar et al., 2007; Rack and Rott, 2004) and widespread supraglacial lake drainage (Banwell et al., 2013; Leeson et al., 2020). Enhanced surface melt likely occurred as zonal winds were +0.33 m/s stronger and a positive SAM anomaly occurred. Positive SAM years with stronger westerlies are not associated with upwelling at the Larsen B and C Ice Shelfs but with warmer air temperatures in combination with warm winds and surface melt (Rack and Rott, 2004). Recent studies found that increases in foehn days in a series of years are related to positive SAM phases and significantly increase surface melt (Leeson et al., 2017; Cape et al., 2015). Interestingly, surface melt was high in the first decade with up to 5 mm w. eq. per day, but still lower than the values recorded during the reference period., The only difference was that the spatial distribution of surface melt changed slightly. Mean air temperature between 1997-2008 increased with +0.16°C (within the uncertainties of the ERA5 data) during summer with no change in winter compared to the reference period. However, in the centre of the ice shelf increases of up to +0.49°C were observed during summer. The observed changes favour supraglacial lake formation as mentioned in the literature. It remains unclear whether the shortly after observed lake drainage is a cause or effect of ice shelf break up (Leeson et al., 2020) and internal glaciological factors might also have played a role due to certain zones of weakness in the ice shelf (Khazendar et al., 2007).

In contrast stands the tabular calving event of iceberg A-68 (~5800 km$^2$) from Larsen C Ice Shelf which accounted for almost the entire ice shelf extent decrease for the I-Ipp basin in 2009-2018 (see Figure 2). The calving event was predictable, as a decade earlier a rift developed from a crevasse field at the Gipps ice rise and propagated further until July 2017 when the iceberg calved (Hogg and Gudmundsson, 2017). In the period of the Larsen C calving event, we could not observe any environmental forcing as lower amounts of melt (-0.2 mm w. eq. per day), slightly stronger zonal winds of +0.1 m/s and decreased summer air temperatures (-0.26°C) occurred compared to 1982-1996. This strengthens the hypothesis of a natural calving event as proposed by Hogg and Gudmundsson (2017). Nevertheless, slight negative thickness changes were observed by Paolo et al. (2015) between 1994 and 2012, which could indicate a future weakening of the Larsen C Ice Shelf. Larsen D forms the only exception of the strong retreating trend along the Antarctic Peninsula. This ice shelf neither experienced melt nor positive trends in zonal winds. Positive thickness changes indicate so far no potential weakening through basal melt (Paolo et al., 2015). The mass balance of the Larsen D to G ice shelves is, depending on the study, positive (Gardner et al., 2018) to slight negative but stillsmaller than the strong negative mass balance of the remaining Antarctic Peninsula (Rignot et al., 2019). Geroge VI and Stange ice shelves were relatively stable during 1997 to 2008 and started to retreat in 2009-2018. Slightly

strengthening westerlies (+0.25 m/s) but almost no melt (0.02 mm w. eq. per day) occurred within the second decade. Summer sea surface temperatures increased by +0.62°C and +0.38°C for George VI south and Stange ice shelves in the second decade compared to the reference period. The retreat rates of both ice shelves started to double the retreat rate during the second decade compared to the first. George VI is not believed to disintegrate rapidly (Holt et al., 2013). But recent developments might require reconsideration of this assumption because calving front retreat, recently detected supraglacial lakes on the ice shelf surface (Dirscherl et al., 2020) and moderate basal melt (Paolo et al., 2015) have occurred.

## 5.2    West Antarctica

Glacier and ice shelf front retreat of the WAIS is clearly dominant at Pine Island Bay and at the biggest Antarctic ice shelves (Ross and Ronne). Ross Ice Shelf lost an area of ~14.000 km$^2$ between 1997 and 2008 which can be completely attributed to tabular iceberg calving of B15-B18 in 2000 and B19 in 2002 (in total ~18.000 km$^2$) (Budge and Long, 2018; MacAyeal et al., 2001). Note that the difference in iceberg size versus retreated area arises from the rough estimation of iceberg area (width multiplied by length provided by the BYU Antarctic iceberg tracking database (Budge and Long, 2018))) as well as the measurement over one decade where re-advance occurred after the calving event. The same applies for Ronne Ice Shelf where the area ( ~12.400 km$^2$) was lost by tabular iceberg calving of A38 and A39 (1998) and A43 and A44 (2000) with an area of ~17.500 km$^2$ (Budge and Long, 2018; MacAyeal et al., 2001; Wuite et al., 2019). Even though the calved icebergs were of notable size (including the largest ever recorded iceberg B15) those calving events were not exceptional (MacAyeal et al., 2001). The Ross West Ice Shelf already experienced such a calving event before 1962, where the front had reached a similar position to that obtained in 2004, after its maximum extent around 1997 (Ferrigno et al., 2007). Also the maximum extent of the Ronne Ice Shelf was reached in 1997 and its minimum extents occurred in 1974 and 2004 (Ferrigno et al., 2005). Measurements of environmental conditions substantiate a natural calving event within the calving cycle of these ice shelves. We could neither observe a reduction of sea ice nor increases in sea surface temperature (0.02 °C) and zonal winds (-0.15 m/s). Strong surface melt (0.5 mm w. eq. per day) occurred solely on the calved area of the Ronne Ice Shelf, but was even higher between 1982 and 1996 at this location. It should be noted that Ronne and Filchner ice shelves might be prone to changing environmental conditions in the future, as recent studies predict that the Ronne Ice Shelf will be affected by increasing basal melt through changing sea ice conditions and wind direction (Darelius et al., 2016).

A completely different setting occurs in the Pine Island Bay where Pine Island and Thwaites glaciers showed strong retreat rates with 40±0.15 km$^2$/yr and 288±0.77km$^2$/yr between 1997 and 2018. The instability and unstoppable retreat of both glaciers has been suggested by several studies (Joughin et al., 2014; Milillo et al., 2019; Parizek et al., 2013; Rignot et al., 2014). The deep marine basin in combination with a retrograde slope creates an increased sensitivity to basal melt by ocean forcing. Our climatological analysis indicates potential upwelling of CDW through strengthening westerlies of +0.28 m/s in 1997-2008 and +0.25 m/s in 2009-2018, compared to the reference period. Additionally, sea surface temperatures increased by +0.28°C at the front of Pine Island but only +0.11°C at Thwaites Glacier from the first to the second decade. Notable changes in sea ice days

or snowmelt were not observed. The sea surface temperature is especially important for the stability of the Thwaites Glacier tongue as it is currently stabilized by persistent fast ice and the recent acceleration of the ice tongue is associated with changing ocean conditions (Miles et al., 2020). It has to be considered that the initial start of destabilization in Pine Island Bay occurred previous to our observation period as mass loss has occured since 1979 (Rignot et al., 2019), new rifting areas were created in the beginning of the 1990s (Bindschadler, 2002; Rignot, 2002) and basal melt by ocean forcing has occurred at least since 1994 (Jacobs et al., 2011). Once marine ice sheet instability is initiated changes in ocean forcing can no longer directly be linked to ice flow and hence calving front position (Christianson et al., 2016). Moreover, as soon as the maximum ice thickness at the grounding line is reached through grounding line retreat on a retrograde bed, the terminus becomes unstable and calving-induced frontal retreat has to be expected, which would no longer be coupled to environmental forced basal melt (Bassis and Walker, 2012). Consequently, it cannot be ruled out that the measured frontal retreat over the last two decades is actually a response to earlier destabilization by ocean forcing. Nevertheless, our observations confirm that Pine Island and Thwaites Glacier are still exposed to ocean forcing.

Besides the fast changing Amundsen Sea Sector, the Bellingshausen Sea Sector at Abbot Ice Shelf is less studied but also vulnerable to CDW forcing (Christie et al., 2016). Just recently, the velocity at the grounding line increased (+20%, 2007-2014), a dynamic thinning signal evolved which is related to ice dynamics and lower snowfall (Chuter et al., 2017) and minor grounding line retreat has occurred (Christie et al., 2016). The recently dynamic thinning signal observed by Chuter et al. (2017) is in line with the glacier front retreat at Abbot Ice Shelf observed since the second decade, where westerlies increased by +0.24 m/s from the first to the second decade, indicating vulnerability to upwelling CDW. Other environmental variables did not change significantly. Along Marie Byrd Land the Getz Ice Shelf experienced strong grounding line retreat between 2003-2008 due to ocean forcing which slowed down afterwards (2010-2015) (Christie et al., 2018). This pattern in ocean forcing is visible in our data as zonal winds reduced by -0.41 m/s from the first to the second decade. Interestingly, this is not apparent in the glacier front movement as glacier front retreat and advance was balanced between 1997 and 2008, but advance reduced from 103 km$^2$/yr to 73 km$^2$/yr in the second decade. The lack of a clear signal in frontal retreat could be explained by several pinning points actioning to stabilize the ice shelf (Christie et al., 2018), as well as the low glacier flow velocity (Mouginot et al., 2019). We propose that those stabilized fronts require longer ocean forcing to respond with frontal retreat. The most stable ice shelves along West Antarctica are the Sulzberger and Nickerson ice shelves because a buffer to ocean forcing is provided by the bathymetry of the continental shelf (Christie et al., 2018). Additionally, they are floating in the cooler Ross Sea (Rignot et al., 2013). No increase in the measured potential environmental variables was observed over the last two decades in this region which further strengthens the hypothesis that these stable ice shelves are not affected by environmental forcing.

## 5.3 East Antarctica

East Antarctica was long believed to represent the invulnerable part of Antarctica with a stable discharge over the last years (Gardner et al., 2018). Recent mass balance estimations show that East Antarctica has been losing mass since 1979, with an

increasing loss since 1999 (Rignot et al., 2019), however altimeter measurements have recorded an almost stable mass balance for East Antarctica since 1992 (IMBIE, 2018). In contrast to the calving front retreat of West Antarctica and the Antarctic Peninsula, glacier and ice shelf front advance has dominated over the last two decades in East Antarctica. The regional

differences are discussed in the following paragraphs and are shown in Figure 2

In Victoria Land, previous studies could not identify a trend in calving front location change and attributed observed changes to glacier size, terminus type and local topographical settings (Fountain et al., 2017; Lovell et al., 2017). Our analysis strengthens the hypothesis that non-climatic drivers regulated calving front change along Victoria Land because no noticeable changes in sea ice, wind direction, snowmelt and ocean surface temperatures were measured between 1997 and 2018, compared

to the reference time period. The only noticeable change was the increase of summer (+ 1.1°C) and winter (+0.71°C) air temperatures over the Mariner and Tucker glaciers during the second decade compared to 1982-1996, which did not influence the slight advance of Mariner Glacier and the stable front position of the Tucker Glacier.

Oats and George V Land have experienced a negative mass balance since 1999 (Rignot et al., 2019) but have not shown obvious trends in calving front position change. The calving front position has been of fronts was balanced over the last two

decades with switches between advance and retreat phases. A previous study by Lovell et al. (2017) noticed the potential influence of a decreased sea ice cover on the terminus position but as we did not measure a significant change in sea ice nor wind direction, sea surface temperature nor melt, the observed frontal changes cannot be connected to environmental forcing. Winter air temperature cooled by up to -1°C and summer air temperature by up to -0.49°C along the coast but both, frontal advance and retreat occurred. The only noticeable retreat occurred at Mertz Glacier tongue which was caused by mechanical

forcing due to an iceberg collision in 2010 (Giles, 2017).

The mass loss of East Antarctica is dominated by the loss in Wilkes Land being the only basin in EAIS loosing mass since 1979 (Rignot et al., 2019) where the majority of glaciers retreated between 2000 and 2012. This contrasting retreat compared to the rest of East Antarctica was linked to environmental forcing by a decrease in sea ice days (Miles et al., 2016). For Wilkes Land, we observed a simultaneous retreat in glacier fronts between 2009 and 2018, but could not link it to a decrease in sea

ice days.

This might be due to the fact that during 1997-2008 sea ice days were only ~ 5 days less than the reference mean, as compared to the ~34 days less between 2000 and 2012, as reported by Miles et al. (2016). This suggests that only extreme reductions in sea ice can be related to glacier retreat, implying that sea ice alone cannot be the only explanation for retreat along Wilkes Land. We propose that upwelling CDW weakened the glaciers by basal melt, as a very strong positive tendency in zonal wind

(with up to +0.44 m/s) being observed in the second (but not in the first) decade compared to the reference mean. All other measured variables did not change significantly over the observation time period.

A strong retreat within the second decade occurred also at Glenzer Glacier next to Wilkes Land. The retreat is not yet mentioned in literature but satellite imagery reveals that an important pinning point was lost in 2004 and a rift started to propagate and lead to a break up event in April 2020. We suggest that the initial destabilization and loss of the pinning point was caused by

an increase in zonal winds (+0.32 m/s) as well as enhanced surface melt (up to +0.4 mm w. eq. /day) during the first decade.

Increased zonal winds persisted during the second decade (+1.23 m/s) indicating further ocean forcing. Nevertheless, the more recent calving event in April 2020 occurred due to a rift that developed over several years. If environmental drivers played a role or whether the rift developed due to ice flow acceleration initiated by the previous lost pinning point requires further investigation.

The fronts of Shackleton and West ice shelves advanced within the last two decades. Both ice shelves are stabilized by several pinning points. Recent studies found that supraglacial lake formation on the Shackleton Ice Shelf is closely connected to short lived high magnitude snowmelt events (Arthur et al., 2020). So far, the supraglacial lake coverage on the ice shelf is still small (< 1%) but it is proposed that widespread ponding and hydrofracture is likely to increase (Arthur et al., 2020). For both ice shelves we observed warmer air temperatures within the first, and colder during the second decade (compared to the reference

period). Snowmelt increased only locally along the margins of both ice shelves (up to +0.3 mm/day). The strongest environmental forcing we observed within the second decade were zonal winds that strengthened by +0.97 m/s and +1.2 m/s for the Shackleton and West ice shelves, respectively. The only indicator that weakening by basal melt occurred indicates the collapse of iceberg D-15. This iceberg calved in 1992 and remained there by re-attaching to the West Ice Shelf but became unstable and split-up in 2015 which has been related to changes in the ocean environment (Walker et al., 2019). Shackleton

shows first signs of instability as ice flow velocities increased by 43 % between 1957-2016 (Rignot et al., 2019). Further investigations are necessary to prove that the measured enhanced zonal winds caused enhanced basal melt at both ice shelves or if other factors caused acceleration and the break-up of the iceberg.

The front of the Amery Ice Shelf gradually advanced between 1997 and 2018. The environmental conditions changed with strengthening westerlies by + 0.26 m/s in the second decade and increased snowmelt (+ 0.23 mm w. eq. per day) on the northern

part of Amery ("Loose Tooth" region close to MacKenzie Bay, see Figure 1) over the last two decades, but a decrease in snowmelt (-0.1 mm w. eq. per day) in the southern region. The stability of the ice shelf is confirmed by velocity measurements where no speed-up occurred since the 1970s (Rignot et al., 2019) and basal melt rates are low (Paolo et al., 2015), suggesting that the westerlies did not strengthen enough to cause upwelling CDW. In 2019, the tabular iceberg D-28 calved from the Amery Ice Shelf. This calving front retreat was predicted by Fricker et al. (2002) based on the observed rift propagation and

regular calving cycle of the ice shelf. Still, it remains unclear if the observed increase in surface melt influenced the rift propagation in the Loose Tooth region. Prior to 2006, the rift propagation was not influenced by environmental forcing but Bassis et al. (2008) did not exclude the potential influence of surface melt if the mean air temperature would raise above zero. The area loss of Enderby Land is mostly attributed to the changes in Shirase Bay. Here, very contrasting changes appear where the Prince Harald Ice Shelf retreats and the Shirase Glacier advances and vice versa. This was already mentioned by Jezek

(2002). The only change in environmental conditions in this area occurred between 1997 and 2008 with an up to +1 mm w. e. melt per day over Prince Edward Ice Shelf compared to the reference period. For Shirase Glacier no increases in melt were observed. In Shirase Bay, the specific glacier morphology and bed topography have been suggested to influence the calving regime, rather than environmental factors (Nakamura et al., 2007).

Dronning Maud Land is dominated by advance over the entire study period. Historical records reveal that along Dronning Maud Land phases of retreat and advance are typical, with larger calving events occurring between 1963 and the 1970s. Retreat continued until the 1990s followed by a phase of advance (Miles et al., 2016; Kim et al., 2001). Goel et al. (2020) found that 95 % of the fronts in Dronning Maud Land advanced between 2009 and 2014. Mass balance estimates also confirm phases fluctuating between mass loss and mass gain since 1979, with no evidence for long term change (Rignot et al., 2019). In addition basal melt rate estimates confirm the relatively stable state of the ice shelves (Rignot et al., 2013; Paolo et al., 2015). Additional stability is given by the distribution of ice rises and rumples and vulnerability to future unpinning exists for the Baudouin and Brunt ice shelves (see Figure 2) (Goel et al., 2020). The environmental conditions along Dronning Maud Land changed only little in regards to sea ice days (-5 days in the first decade) and constant zonal winds between Shirase Bay and the Fimbul Ice Shelf, and more negative winds (~ 0.35 m/s) between Jelbart and Filchner ice shelves (1997-2018 compared to the reference mean). The only exceptional change was observed for air temperature which increased between Lazarev and Filchner ice shelves between +0.6-1.6 °C in summer and +0.9-1.8 °C in winter within the second decade compared to the reference mean. Despite this remarkable increase in air temperature ice shelf advance persisted as mean air temperatures were far below the freezing point and surface melt did not increase.

## 5.4    A Circum-Antarctic Perspective

The above examples illustrate that environmental conditions have changed non-uniformly along the Antarctic coastline over the last two decades. Strengthening westerlies occurred along with frontal retreat at the Western Antarctic Peninsula, Larsen B Ice Shelf, Pine Island Bay, Abbot Ice Shelf and Wilkes Land, but advance has occurred at the Shackleton and West ice shelves. This pattern can be due to several reasons. Though strengthening zonal winds are a proxy for upwelling CDW (Spence et al., 2014) further studies are needed to assess the relationship between the magnitude of strengthening westerlies and the amount of basal melt resulting from that. Only a small increase in westerlies could be enough to increase basal melt considerably, because basal melt has a nonlinear relationship with ocean temperature and can quadruple with only slight ocean warming (Jenkins et al., 2018). Additionally, the forcing by zonal winds is less strong in areas where CDW upwelling is less efficient, due to strong tidal forcing (Hazel and Stewart, 2019; Stewart et al., 2018) and the local topography regulates the amount of CDW accessing the cavities of an ice shelf (Dutrieux et al., 2014). Moreover, the duration of required environmental forcing to cause destabilization of ice shelves solely by basal melt very likely exceeds the time scales addressed in this study. In addition, Shackleton and West ice shelves only experienced potential forcing by one driver but it was found that the forcing by several environmental drivers had the greatest impact on calving front retreat of tidewater glaciers in Greenland (Cowton et al., 2018; Howat et al., 2008), which likely also applies for Antarctic ice shelves and glacier tongues. Further studies are necessary to assess the amount of basal melt for individual ice shelves as a result of strengthening westerlies.

Our study found a strong link between increased sea surface temperature and the occurrence of calving front retreat. The measured increases in sea surface temperature were between +0.15 to +0.6°C per decade which is still below the trend uncertainty stability of 0.01°Cper year (Embury, 2019). The influences of warmer sea surface temperatures are twofold.

Warmer sea surface temperatures lead to the melt of fast ice which destabilizes glacier margins (Larour, 2004), and could result in retreat. Another explanation proposes that warm surface water increases melt at the waterline which creates an overhanging ice cliff that is more likely to collapse (Mosbeux et al., 2020). Our results suggest, that increased sea surface

temperatures weakened the margins in the Bellingshausen and Amundsen Sea Sector, where highest increases in sea surface temperature were observed. Further destabilization could arise through the presence of marginal weakening rifts that become unstable (Lipovsky, 2020). Marginal weakening can also arise from a decrease in sea ice days which occurred along the Western Antarctic Peninsula, the Bellingshausen and Amundsen Sea Sector and in Wilkes Land. Decreasing sea ice days occurred in conjunction with glacier/ice shelf retreat if the average reduction per year was over ~10 days or more within the

decade. This might become important in the future. Recently, the turning point of an increasing Antarctic sea ice extent was reached (Parkinson, 2019; Ludescher et al., 2019).

Relative changes in mean air temperature could not be identified as a direct driver for calving front retreat, even though increases of up to +2°C per decade were measured in some coastal areas (e.g. Dronning Maud Land, Victoria Land) which is beyond the uncertainty of the ERA5 air temperature data. This suggests that mean air temperature over a decade is not an

appropriate way to assess the effect of air temperature on calving front change. This is because the relationship between air temperature driven surface melt and hydrofracture is known to destabilize ice shelves and can cause glacier front retreat (Arthur et al., 2020; Banwell et al., 2013; Leeson et al., 2020). More suitable would be the assessment of the amount of positive degree days and temperature extreme events that directly influence surface melt on the ice shelf, with more accurate and higher-resolution melt data. For example, catchment wide melt and the resulting runoff had a higher impact on the retreat of

Greenlandic glaciers than local air temperatures (Cowton et al., 2018).

Howat et al. (2008) found that peak events in air temperature and sea surface temperature can initialize glacier retreat. This study found that the absolute amount of snowmelt correlated significantly with glacier and ice shelf retreat but not relative changes in snowmelt. The detailed comparison of snowmelt with calving front retreat revealed strong melt during the decade of the break up event (e.g. Larsen B, Wilkins ice shelves). Nevertheless, snowmelt was often higher during the reference period

than during the actual calving event which might indicate that longer phases of melt weaken the ice shelf over a longer time period (Leeson et al., 2017), and the intensity of surface melt events are of higher importance than just the presence of melt (Arthur et al., 2020). Moreover, the low resolution of melt data did not always match the actual calving front. We assume that the use of a not up to date coastline product (e.g. at Larsen B) caused the inaccuracies over the ice shelf margins. Additionally, no information on the accuracy of the snowmelt product is provided which makes the considered reanalysis data prone to

potentially large biases. We suggest a more detailed study on glacier and ice shelf front change in combination with accurate high resolution surface melt data (e.g. satellite-based surface melt estimates as published by Trusel et al. (2013)) to capture all necessary surface hydrology processes including peak events of snowmelt. For example, it would be insightful to quantify the effect of ponding on ice shelf vulnerability rather than calculating the exact amount of melt (Joughin and Alley, 2011). Moreover, the hydrological system of the ice shelf has to be considered as river formation can act as a stabilizing mechanism

by exporting meltwater to the ocean (Banwell, 2017),and hydrological processes such as lake drainage that occurred prior to

the break up event of Larsen B Ice Shelf have to be assessed further (Leeson et al., 2020). These complex surface hydrological processes require much more detailed studies that consider the combination of factors mentioned above and not solely focusing on mean surface melt.

The identification of environmental drivers for calving front retreat presented a major challenge and some questions remain unresolved due to the limitations of this study. The connection between atmospheric forcing and calving front retreat could not be entirely revealed as average surface melt and air temperatures did not allow for an analysis regarding threshold behavior (e.g. above zero-degree days), peak events and surface hydrological processes. Moreover, once ice shelf destabilization is initiated the response in frontal change may no longer linearly be connected to forcing and become decoupled from external forcings as known for the marine ice sheet instability (Feldmann and Levermann, 2015; Joughin et al., 2014; Robel et al., 2019). As a result, the natural calving cycle and responses to environmental forcing may prevail on longer time scales than the ones addressed in this study. Those shortcomings were addressed by discussing the observed environmental changes of this study with information from previous studies allowing the identification of key drivers for calving front retreat being not solely environmentally driven but a combined result of complex interactions between internal ice dynamics, geometry, external mechanical forcing and environmental drivers.

For future assessments of environmentally-driven calving front change, it has to be considered that the environmental drivers with the potential to drive calving front retreat are closely connected to SAM and reinforced during positive SAM years. Extreme peaks in positive SAM can be connected to ice shelf disintegration, as has been shown for the Wordie Ice Shelf (Walker and Gardner, 2017), Larsen B Ice Shelf, as well as the first break-up event of Wilkins Ice Shelf. A positive SAM increases sea surface temperatures and air temperatures along the Antarctic Peninsula and West Antarctica. It also influences the sea ice cover, snowmelt and the foehn effect, as well as enhancing westerly winds along the Antarctic coastline (Verdy et al., 2006; Kwok and Comiso, 2002; Marshall, 2007; Tedesco and Monaghan, 2009; Cape et al., 2015). Glacier and ice shelf retreat forced by environmental drivers will be even more likely in the future as rising greenhouse gases and ozone depletion will cause more positive phases of SAM (Paeth and Pollinger, 2010; Wang et al., 2014), further strengthening the identified environmental forces on calving front retreat.

## 6    Conclusion

For the first time, we present a circum-Antarctic record of calving front changes over the last two decades. Overall, the extent of the Antarctic Ice Sheet decreased -29,618±1,193 km$^2$ between 1997 and 2008, and gained an area of +7,108±1,029 km$^2$ between 2009-2018. Glacier and ice shelf front retreat concentrated along the Antarctic Peninsula and West Antarctica. The only East Antarctic coastal sector experiencing simultaneous calving front retreat of several glaciers was Wilkes Land in 2009-2018. The largest proportion of calving originated from Ross West and Ronne ice shelves being responsible for 75 % of the West Antarctic loss during 1997 and 2008, but this loss was found not to be exceptional as similar minimum extents of the Ross and Ronne ice shelves occurred in the 1960s and 1970s, respectively. Decreasing sea ice days, strengthening westerlies,

intense snowmelt and increasing sea surface temperatures were identified as enabling driving forces for glacier and ice shelf front retreat along the Antarctic Peninsula, West Antarctica and Wilkes Land. In contrast, changes in mean air temperature were not an appropriate measure for atmospheric forcing and rather extreme temperature events, threshold behaviour and surface hydrological processes should be considered for future analysis. Further studies should assess whether record-high air temperatures can also trigger glacier/ice shelf retreat as it has been observed for Greenland. Snowmelt was found to be a strong driver of calving front retreat at the Antarctic Peninsula (up to 5 mm w. eq. per day) but more accurate data on surface melt and surface hydrology is needed to assess the influence of melt in more detail. Increased sea surface temperatures (of up to +0.62°C) were observed along the Bellingshausen and Amundsen Sea Sectors weakening the glacier margins. Between 1997 and 2018, sea ice days decreased the most along the Western Antarctic Peninsula, by up to -25 days, the Bellingshausen Sea Sector (-10 days) and Amundsen Sea Sector (-5 days). Strengthening westerlies affected ice shelves along the Western Antarctic Peninsula (up to +0.54 m/s) and West Antarctica (+0.28 to +0.41 m/s) but also in East Antarctica, particularly in Wilkes (+0.44 m/s) and Queen Mary Land (up to +1.23 m/s). Despite those changes in environmental conditions the influence of internal ice dynamics, geometry and external mechanical forcings play a crucial role and must not be neglected when assessing drivers of calving front retreat.

The assessed environmental drivers are closely connected to positive phases of SAM which occurred over the last two decades. Rising $CO_2$-emissions and ozone depletion will further enhance positive phases of SAM putting additional pressure on glaciers and ice shelves. To better asses the vulnerability of glaciers and ice shelves in the future, it is essential to better understand surface melt, ponding and surface runoff processes that are impacting calving front retreat to an unknown extent. Equally important is the understanding of changing wind conditions and their impact on upwelling CDW and basal melt. Another interesting issue related to glacier and ice shelf changes across Antarctica pertains to their specific time scales and response times. A shortcoming of our analysis is the restriction to three snapshots of coastline data that has limited the scope of this study. Although our study has revealed that the Antarctic cryosphere is subject to tremendous changes in recent decades, it is conceivable that our results are at least partly biased by strong inter-annual variability of all considered climate and cryospheric variables. In addition, natural cycles and the effects of man-made global warming may prevail on longer time scales than the ones we selected for this study on the basis of reasonable data coverage.

**Data availability**

The Antarctic coastline 2018 produced within this study is available at the EOC Geoservice of the German Aerospace Center (download.geoservice.dlr.de/icelines/files/). The supplementary material provides calving front changes for each assessed glacier/ice shelf in m/yr and km$^2$/yr (see Table S1).

The Radarsat (1997) and MODIS (2009) coastlines are available at nsidc.org/data/NSIDC-0103/versions/2 (Jezek et al., 2013) and nsidc.org/data/NSIDC-0593/versions/1 (Haran et al., 2014), respectively. Downloaded ERA5 data is available at the Copernicus Climate Data Store from cds.climate.copernicus.eu (Copernicus Climate Change Service, 2019a, b) and at the

Ocean and Sea Ice Satellite Application Facility from www.osi-saf.org/?q=content/sea-ice-products (OSI SAF, 2017). The BYU (Brigham Young University) Antarctic iceberg tracking database can be accessed from scp.byu.edu/data/iceberg/database1.html (Budge and Long, 2018). The LIMA Landsat Mosaic was downloaded from the USGS webpage lima.usgs.gov/fullcontinent.php (Bindschadler et al., 2008). The TanDEM-X PolarDEM 90 m was generated as described in Wessel et al. (2021) and is freely available at download.geoservice.dlr.de/TDM_POLARDEM90/ANTARCTICA/.

## Supplement

The supplement related to this article is available online at: XYZ.

## Author contributions

CB designed the study, conducted the analysis and wrote the original draft of the manuscript. CK and AD assisted in the study design. CB, CK, AD, CHK and HP contributed to the discussion of the results and were involved in editing the manuscript.

## Competing interests

The authors declare that they have no conflict of interest.

## Acknowledgements

We thank the Copernicus Climate Data Store for providing the ERA5 climate data and the Ocean and Sea Ice Satellite Application Facility (OSI SAF) for providing the sea ice concentration datasets. In addition, we thank the European Union Copernicus program providing Sentinel-1 data accessible via NASA Alaska Satellite Facility (ASF) and G. J. Marshall for granting open access to the Southern Annular Mode (SAM) Index. Finally, we would like to thank the editor, especially Eleri Evans, Brad Lipovsky and one anonymous reviewer for their comprehensive comments and suggestions on our manuscript.

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
