# Peer review of "Environmental Drivers of Circum-Antarctic Glacier and Ice Shelf Front Retreat over the Last Two Decades"

_The Cryosphere, 2020_

## Short Comment (SC1) · 5 Oct 2020

This paper is unusually thorough and represents a major contribution to the field. I will certainly build off of these results in my own work.

I have a few minor comments. It'd be interesting to have the information in Table 2 presented in m/y by taking the areal rate km^2/yr and dividing by the ice front length. This would be useful for modelers who often calculate "calving rates" in m/yr.

Second, it would be interesting to explain in a bit more detail how observed retreat rates are related to major tabular calving events. See list here:

[Figure]

https://www.scp.byu.edu/data/iceberg/. For example, consider B15, the biggest known iceberg. How much of the 868±1.3 kmˆ2/yr (= 9548 kmˆ2 over 11 yr) number for Ep-F in Table 2 is due to the calving event that created iceberg B15? Let's see. . . it was about 200km x 50km. So that's 10,000 kmˆ2. In other words, the entire amount of retreat. This might be worth noting.

Third, it would be interesting to see a bit more discussion of the relationship between calving physics and the observed retreat rates. The authors cite the paper by Larour 2004, Yu et al., 2017, and Mosbeux et al., 2020, which is a great start! But there are many other ways that surface and basal melting have been proposed to relate to calving. Consider these papers, for example:

Bassis, Jeremy N., and C. C. Walker. "Upper and lower limits on the stability of calving glaciers from the yield strength envelope of ice." Proceedings of the Royal Society A: Mathematical, Physical and Engineering Sciences 468.2140 (2012): 913-931.

Lipovsky, Bradley Paul. "Ice shelf rift propagation: stability, three-dimensional effects, and the role of marginal weakening." The Cryosphere 14.5 (2020): 1673-1683.

Banwell, Alison F., Douglas R. MacAyeal, and Olga V. Sergienko. "Breakup of the Larsen B Ice Shelf triggered by chain reaction drainage of supraglacial lakes." Geophysical Research Letters 40.22 (2013): 5872-5876.

Overall, I find this paper to be exceptional in its quality and depth. Well done!

---

## Referee Comment (RC1) · Eleri Evans (Referee) · 19 Oct 2020

**General Comments**

This paper describes the production of a calving front change dataset which will surely be of significant interest to the Antarctic glaciological community. In addition to manually correcting and adjusting previously published coastline datasets this work has also produced a more recent Antarctic coastline using Sentinel-1 imagery (from 2018) and CNN techniques. The resulting terminus position change data will be invaluable to many studies investigating ice shelf and glacier tongue behaviour. Furthermore, the novel use of recently produced reanalysis (ERA5) outputs, in con-

junction with other environmental data, has allowed a unique investigation into the key environmental drivers that have influenced ice shelf and glacier calving front behaviour.

I have some concerns with wording and sentence/paragraph structure used within the Discussion, as environmental parameters have been singled out to be the sole driver of calving events, and hence frontal retreat, with a complete neglect of the glaciological forcings involved. However, I think with careful rewording and strengthening of key arguments some really robust and significant findings can be presented here. I urge the authors not to be discouraged by the length of my review (particularly the technical comments) as the comments are intended mainly to help with grammar that will hopefully help improve the manuscript.

**Specific Comments**

The placement of Figure 1 is unusual, though it is referred to within the introductory text it contains results from the analysis performed in this work. Therefore, I suggest that this figure may be better suited to the Results section.

I find the term 'the long term mean (1982-1996)' to be confusing. How was this mean derived and how is it long term? Shouldn't the long term mean be from 1982-2018 rather than referring to a previous epoch (1982-1996)? Using different terminology instead of 'long term mean' will likely remove this confusion.

The first few sentences of the Zonal wind Results section (line 280), that introduce SAM, read more like they would be better suited to the Zonal wind section of Section 2.2. In addition, the results for SAM (Figure 9) are included in the Discussion. I strongly suggest moving the SAM section from the Discussion to the Results section following the zonal wind results, as this will clarify the linkage between zonal winds

and trends in the SAM.

It is imperative that there is careful wording used (particularly in sections of the Discussion) regarding the correlation between changes in the environmental variables and the pattern of calving front retreat, as is correctly mentioned at the start of the Discussion; correlation does not necessarily mean causation. The abstract uses appropriate language, e.g. 'enabling factors' which suggests that the environmental variables may be involved in destabilising ice shelves prior to calving, such as enhancing surface melt rates or driving ice shelf thinning, but they are not necessarily acting alone, as there are likely to be glaciological factors involved in the calving events. However, many of the paragraphs in the Discussion become misleading given paragraph structure, the words used and the lack of consideration of glaciological processes (particularly rift development). The impact of neglecting glaciological factors becomes clear in the argument regarding air temperature and calving front retreat (lines 329-332). Suggesting that the air temperature parameter is not a key environmental driver because there is a retreat of EAIS ice shelves under cooler air temperatures is very misleading. It allows neglect of the very important relationship between air temperature driven surface melt and hydrofracture, that has been found to be a part of disintegration style calving events elsewhere in Antarctica. Another key example is regarding the Amery Ice Shelf (lines 453 – 457), the wording and sentence structure implies that surface melt was involved in the recent calving event of 2019, '...the part affected by increased surface melt broke off in 2019...'. This is a very misleading sentence as it completely ignores that rift propagation was the driving factor involved in this calving event.

I suggest restructuring the paragraphs in the Discussion that focus on individual ice shelves or particular calving styles, to begin with mention of the complex interactions between the glaciological forcings and the environmental forcings that previous studies
have identified to be involved in the calving events. Then follow this with the supporting evidence from the work performed here for the involvement of the environmental forcings in the observed frontal retreat. Including mention of the specific glaciological forcings as well as referencing studies that have looked at the glaciological drivers of calving will reduce confusion surrounding the key drivers, strengthen the arguments regarding the important influence of the environmental forcings and allow key environmental variables to be identified for future change analysis.

**Technical Comments**

Line 1. As only environmental drivers were evaluated with regard to calving front retreat I suggest the title should be amended to reflect this, e.g. 'Environmental Drivers of Circum-Antarctic...'
Line 11. Replace 'being' with 'that are'.
Line 14. Antarctic ice sheet should be capitalised so it is Antarctic Ice Sheet, this needs to be amended throughout the manuscript.
Line 15. Try to avoid informal language throughout the manuscript; such as shrank (replace with decreased by), comes along (replace with occurs in conjunction, line 17), actually (remove, line 170), hotspots (amend, line 199), slightly (replace with specific values, line 213).
Line 26. 'Shelfs' should be 'shelves', suggest restructuring this sentence so that it reads '...the fronts of ice shelves and glaciers...' also check the rest of the manuscript for this plural issue.
Line 30. Calving front retreat will only increase ice discharge if ice that has high buttressing values is removed, so it would be beneficial to clarify this point and add references.
Line 32. Is it supposed to be a single decade or several decades? If several, please add the number.
Line 34. Suggest creating a new sentence at 'Since' and rewording e.g. 'Since the

1990s altimetry measurements have shown a small gain...'.

Line 35. Restructure e.g. 'However, a strong mass loss trend of -47+/- Gt/yr (1989-2017) is calculated using the mass budget method (Rignot et al., 2019)'.

Line 37. Watch out for the use of 'at' instead of 'in' or 'of' throughout the manuscript, it should read the ice shelves of or in a region depending on the context.

Line 39. Make 'Glacier' a plural and add 'the' before 'atmosphere' and before 'ocean'. Also add 'are' following 'hence'.

Line 41. 'natural cycle of decay and growth'. This is a key point as it relates to the glaciological forcings that I mentioned in the Specific Comments. I suggest expanding what you mean by the decay and growth cycle and how glaciological parameters fit into this. That way this paragraph can introduce the relationship between the glaciological parameters and the environmental forcings.

Line 44. Replace 'to' with 'with' before 'atmospheric warming'.

Line 45. Is ocean forcing the main driver or one of the key drivers involved?

Line 48. The upwelling doesn't specifically increase ocean temperatures, but rather allows the warmer ocean waters to reach the base of the ice shelves, so I suggest restructuring this sentence.

Line 50. Avoid using 'Especially' throughout the manuscript, particularly when starting a sentence, and restructure this sentence e.g. 'Basal melt, when combined with a retrograde bed (refs), has lead to a retreat of the grounding line followed by increased ice discharge (refs).' Maybe add for which ice shelves/glaciers this has occurred.

Line 52-53. 'In contrast...Miles et al 2016'. This sentences needs better incorporation into the paragraph and explanation of what sea ice days are is also needed. I also suggest possibly making a new paragraph here as the existing one is very large.

Line 84. Re-analysis vs reanalysis on line 87, check for consistency throughout the manuscript, the same applies for in-situ and in situ as well as snowmelt and snow melt, ERA-5 and ERA5.

Line 89. Sentence ending '...former ERA-Interim product', suggest adding references.

Line 103. Sentence ending 'especially during summer', suggest adding references.

Line 105. Sentence beginning 'Overall, the mean absolute...' ending is incomplete, suggest adding 'in zonal wind speed is achieved' after 'variability'.

Line 113. 'sea ice months April through to October,' how have you chosen these months? Suggest adding clarification. Are you referring to fast ice or pack ice or both?

Line 119. Add 'than' in between 'higher' and '15 %'.

Line 127. 'Oct-Mar', watch out for consistency when writing the months, choose either abbreviated or in full and apply throughout the manuscript.

Line 186. Suggest replacing 'composes' with 'is derived from' and adding 'the' in between 'for' and 'two' on Line 187.

Line 191. Suggest replacing 'visualized' with 'shown', also on line 251.

Line 195-196. Suggest changing this sentence to be 'Between 1997 and 2008, the large disintegration events of the Larsen B, Wilkins and Wordie ice shelves resulted in a 37 % higher calving amount from the Antarctic Peninsula, as compared to the amount calved from this region between 2009 and 2018.'

Line 205. Best to avoid starting a sentence with 75%, suggest changing the sentence structure. Also I'm not sure what you mean by Ross West and Ross East, I don't think this is a common naming convention and I'm wondering if the Ross West is the McMurdo Ice Shelf? In addition, Figure 2 shows two Ross West labels.

Line 206. When you have multiple ice shelves it is common to capitalise the name and then have ice shelves in lower case. For example, it would be '...the Ross and Ronne ice shelves'. Also it is common when identifying particular ice shelves in the text to refer to them using 'the' e.g. 'the Amery and Filchner ice shelves'. I suggest applying these changes throughout the manuscript.

Line 224. Suggest replacing 'But' with 'However,'.

Line 228. Replace 'Ice Sheet' with 'ice shelves'.

Line 280. Suggest adding 'the' in between 'In' and 'case'.

---

## Short Comment (SC2) · 2 Nov 2020

We would like to thank Bradley Lipovsky for his helpful comments and insights.

In order to make our results more useful for modelers, we decided to additionally add calving rates in m/yr of major ice shelves and glaciers in a supplementary file.

We had a look on the biggest tabular calving events and compared the area loss to the calculated retreat. Indeed, especially for the large ice shelves like Larsen, Ross and Filchner the amount of tabular ice berg calving almost equals the calculated retreat. This finding is now mentioned in the manuscript.

[Figure]

Thank you very much for the recommended literature. The discussion of our manuscript was revised and gives now deeper insights into the relationship between retreat and calving physics as well as surface and basal melt.

---

## Referee Comment (RC2) · Anonymous Referee #2 · 13 Nov 2020

**Overall comments**

As noted by reviewer Eleri Evans and in the comment by Brad Lipovsky, this study produces a very useful new Antarctic ice-margin dataset and mapping of frontal change which is a valuable addition to the literature. It also provides a useful and interesting look at patterns of change in some of the major ocean and climate parameters around Antarctica over the ERA5 period/last ~two decades, on a large scale and in a manner that is consistent and compelling. As the core results of this study, these are strong contributions (I suggest some minor improvements/clarifications below).

The correlation analysis between the frontal changes and climate/ocean parameters is more challenging because the response of ice shelves and glacier fronts to forcing is so markedly non-linear. Thresholds in the response to forcing are common, as are instabilities in which, by definition, the shelf/glacier behaviour becomes divorced from external forcing. These issues are alluded to in parts of the discussion, but need to be addressed.

The behaviour of ice fronts could be seen as the combined result of:

i)     Externally-forced trends like ice-shelf thinning due to increased basal melting (as in the Amundsen Sea Embayment), or the loss of the surface firn cover due to warmer summers (as on Larsen A and B). The forcing of these trends could potentially be diagnosed through correlation, if the right parameters can be measured for long enough (e.g., ocean temperature at depth, or positive summer surface air temperatures). Note though that the response to these forcings is not necessarily linear due to feedback. For example a reduction of albedo as a shelf surface melts acts as a positive feedback, enhancing the sensitivity of melt to shortwave radiation.

ii)    Externally-forced shocks superimposed on these trends, like an exceptionally warm summer (as on Larsen B). These would not be readily captured by decadal climate means and would not necessarily have the same effect on all shelves/fronts, so would be difficult to correlate to frontal change.

       Threshold behaviour could be very important for i) and ii) – e.g., the difference between a summer surface temperature staying just below freezing or just above is profound, with the latter producing meltwater and rapidly densifying the firn. Rapid retreat or full shelf collapse could be triggered a slightly larger than normal retreat of a shelf front that happens to take it back behind a compressive arch of forces (e.g., Larsen A).

iii)   Internal ice dynamics like the calving cycle, with a long, slow advance followed by an abrupt calving event, controlled by the evolving stress field and existing damage to the ice. This is largely unrelated to external forcing. For some shelves with long cycles, these may not be well sampled even by decades-long observations. Calving-cycle events can be large, dominating the statistics of frontal change.

iv)    Unstable dynamic response to trends and shocks. Once initiated, thinning, acceleration, damage and retreat of shelves like PIG and Thwaites may run indefinitely, strongly controlled by evolving ice stress and damage, and only somewhat modulated by the sort of environmental parameters studied here. A marine ice sheet instability (MISI) could, for example, be triggered after several decades of ocean-driven shelf thinning and perhaps some shelf retreat, but once initiated could even drive an ice shelf readvance as ice dynamics took over, regardless of the external forcing. Similarly, a marine ice cliff instability

(MICI) could be triggered by an initial external forcing but then progress into a runaway retreat regardless of what happens to that forcing.

I'd expect that these non-linearities, thresholds and instabilities mean that ice front changes are not likely to correlate well with external forcing….and yet…there are some signals there in the correlation results. And it does seem reasonable that a sustained forcing, like decreased sea ice cover, acting over a large area could drive a coherent signal of change at multiple independent ice fronts, particularly where their calving is dominated by frequent production of multiple small bergs rather than rare, large tabular bergs. The forcings are important because they drive the trends and provide the shocks, but they might be difficult to untangle when looking at all of the ice fronts.

**General suggestions**

Given the above, I suggest that instead of seeking to explain ALL ice-front changes through correlations with external forcing, the discussion section is re-oriented towards addressing the questions:

i)      are ANY of these ice-front changes externally forced? (i.e., which ones can be distinguished with confidence from noise, internal dynamics and dynamic instabilities?)
ii)     then for these fronts where forcing is detectable, which forcings have mattered most?

To do this, perhaps choose subsets of the coast with numerous relatively small, independent ice fronts that are not experiencing major dynamic thinning, are not dominated by rare, major calving events at one or two shelves, and are not still responding to shelf collapses from several decades ago (like Wordie and various other AP coasts probably were). Look for correlations that are spatially coherent on the same scale as the forcing patterns, i.e., affect multiple neighbouring shelves/fronts simultaneously. Consider extracting statistics on decadally-extreme forcing events rather than just decadal means. Consider focussing on beyond-threshold parameters like summer-air-temperatures-above-freezing (or positive degree days) rather than all temperatures. Consider calculating these temperatures only at very low altitude (e.g., <200 m or as appropriate) to focus on the shelves and ice fronts themselves - the 100 km landward buffer currently used will inevitably bias the shelf/front temperatures low, and this could be important. The bias will be particularly big for small, fringing shelves with relatively steep ice sheet inland.

While this involves some extra analysis, I think that you already have the datasets to focus in on these questions.

**Specific suggestions/questions**

Section 3.1 on coastline detection – physically what aspects of the HH/HV signal distinguish the 'ocean' and 'land' classes? i.e., why is it desirable to have HV as well as HH? A contrast in volume-scatter from the land ice and sea ice?

What decisions did you make in defining messy fronts like the collapsing Thwaites Glacier/Iceberg tongue?

Line 142: why use winter scenes rather than summer when open water is more likely?

Section 3.1: can you give more detail on the uncertainty assessment? The total uncertainties of ±29 and ±144 sq km given in the abstract seem exceptionally small.

Line 167: how did you define the 30 'stable' areas used for quality control?

Line 172: do these uncertainties have a sign or are they ±? Is there a tendency towards biasing the fronts too far seaward, because sea ice/melange is sometimes present on the seaward side, and sometimes mistaken for 'land ice' (but not the other way around)?

Line 192: what do these '% of total area' mean? i.e., the 'total area' of what? Do you mean 'total coastline length' instead of area?

Figure 2: I'm confused by the pie chart sizes. What does the size of the pie charts indicate when they include both positive and negative area changes as segments of the pie? e.g., for the Ronne pie in the top panel, what was the advance rate from 1997-2018? Is the size the net area change, which can either be positive or negative?
Ross East label is missing.

Line 231: Do you mean "by 1°C" rather than "of 1°C"?

Figure 6: Larsen B is showing up as having a big increase in snowmelt, but in fact it had by then collapsed.

Line 299: what does "the percentage of retreat/advance within each glacier/ice shelf basin" mean?

Line 376: in contrast, Larsen C has not broken up. It did have a big calving event though.

---

## Author Comment (AC1) · 22 Dec 2020

Response to Referee #1 Eleri Evans

We would like to thank Eleri Evans for the very detailed review and her comprehensive comments on language. The very helpful comments on wording and sentence structure were included to provide clarification and improve the manuscript. Additionally, we included the suggestions regarding the discussion to clarify the difference between environmental and glaciological drivers. We hope the rewording, stressing of key arguments and mentioning of study limitations helped to improve the manuscript significantly. Please find the answers on specific and technical comments below in blue color. The improved manuscript containing the described changes (highlighted with the track changes function) will be provided after we have received the final feedback from the editor.

**General Comments**

This paper describes the production of a calving front change dataset which will surely be of significant interest to the Antarctic glaciological community. In addition to manually correcting and adjusting previously published coastline datasets this work has also produced a more recent Antarctic coastline using Sentinel-1 imagery (from 2018) and CNN techniques. The resulting terminus position change data will be invaluable to many studies investigating ice shelf and glacier tongue behaviour. Furthermore, the novel use of recently produced reanalysis (ERA5) outputs, in con-junction with other environmental data, has allowed a unique investigation into the key environmental drivers that have influenced ice shelf and glacier calving front behaviour. I have some concerns with wording and sentence/paragraph structure used within the Discussion, as environmental parameters have been singled out to be the sole driver of calving events, and hence frontal retreat, with a complete neglect of the glaciological forcings involved. However, I think with careful rewording and strengthening of key arguments some really robust and significant findings can be presented here. I urge the authors not to be discouraged by the length of my review (particularly the technical comments) as the comments are intended mainly to help with grammar that will hopefully help improve the manuscript.

We appreciate the very detailed review and gladly added the comments on grammar to improve our manuscript. Additionally, we tried to give a more holistic view on drivers of calving front change and emphasized the importance of glaciological drivers within our re-structured and improved discussion.

**Specific comments:**

The placement of Figure 1 is unusual, though it is referred to within the introductory text it contains results from the analysis performed in this work. Therefore, I suggest that this figure may be better suited to the Results section.

You are completely right that Figure 1 already includes results of the calving front change analysis. Nevertheless, we decided to put this Figure right at the beginning to provide an overview of locations mentioned in the paper. This should help readers not familiar with Antarctica to locate important coastline sections and glaciers. Therefore, we would like to keep Figure 1 at the beginning of the manuscript and hope you can accept this decision.

I find the term 'the long term mean (1982-1996)' to be confusing. How was this mean derived and how is it long term? Shouldn't the long term mean be from 1982-2018 rather than referring to a previous epoch (1982-1996)? Using different terminology instead of 'long term mean' will likely remove this confusion.

Thank you very much for drawing attention on the misleading term "long-term". To avoid confusion, we decided to re-name the epoch from 1982-1996 to "reference" period.

The first few sentences of the Zonal wind Results section (line 280), that introduce SAM, read more like they would be better suited to the Zonal wind section of Section 2.2. In addition, the results for SAM (Figure 9) are included in the Discussion. I strongly suggest moving the SAM section from the Discussion to the Results section following the zonal wind results, as this will clarify the linkage between zonal winds and trends in the SAM.

*Thank you very much for this comment. We changed the manuscript accordingly and shifted Figure 9 to the results section. Additionally, the description of SAM is now included in Section 2.2.*

It is imperative that there is careful wording used (particularly in sections of the Discussion) regarding the correlation between changes in the environmental variables and the pattern of calving front retreat, as is correctly mentioned at the start of the Discussion; correlation does not necessarily mean causation. The abstract uses appropriate language, e.g. 'enabling factors' which suggests that the environmental variables may be involved in destabilising ice shelves prior to calving, such as enhancing surface melt rates or driving ice shelf thinning, but they are not necessarily acting alone, as there are likely to be glaciological factors involved in the calving events. However, many of the paragraphs in the Discussion become misleading given paragraph structure, the words used and the lack of consideration of glaciological processes (particularly rift development). The impact of neglecting glaciological factors becomes clear in the argument regarding air temperature and calving front retreat (lines 329-332). Suggesting that the air temperature parameter is not a key environmental driver because there is a retreat of EAIS ice shelves under cooler air temperatures is very misleading. It allows neglect of the very important relationship between air temperature driven surface melt and hydrofracture, that has been found to be a part of disintegration style calving events elsewhere in Antarctica. Another key example is regarding the Amery Ice Shelf (lines 453 – 457), the wording and sentence structure implies that surface melt was involved in the recent calving event of 2019, '...the part affected by increased surface melt broke off in 2019...'. This is a very misleading sentence as it completely ignores that rift propagation was the driving factor involved in this calving event.

*We fully agree that careful wording in the discussion is crucial and that fluctuations in glacier and ice shelf front position are the combined result of complex interactions between internal ice dynamics, geometry (e.g. fjord geometry, bed topography) and external forced mechanical (e.g. iceberg collision) and environmental drivers. Therefore, the discussion was improved by more appropriate wording (potential drivers, enabling factors) and a more holistic approach by stressing the importance of glaciological processes including rift development. Additionally, the introduction now includes a more detailed description on the process of calving itself and all factors influencing calving front change.*

*Even though, we could not find a significant correlation between relative changes in mean air temperature we fully agree that a relationship between air temperature driven surface melt exists. As mentioned by reviewer #2 we suppose that rather above freezing temperature days and extreme events in temperature are essential for surface melt and hence the destabilization of ice shelves by hydrofracture. We included this fact in our discussion.*

*„Relative changes in mean air temperature could not be identified as a direct driver for calving front retreat, even though increases of up to 2°C per decade were measured in some coastal areas (e.g. Dronning Maud Land, Victoria Land) which is beyond the uncertainty of the ERA5 air temperature data. This suggests that mean air temperature over a decade is not an appropriate way to assess the effect of air temperature on calving front change because the relationship between air temperature driven surface melt and hydrofracture is known to destabilize ice shelves and can cause glacier front retreat (Arthur et al., 2020; Banwell et al., 2013; Leeson et al.,*

*2020a). More suitable would be the assessment of the amount of positive degree days and temperature extreme events directly influencing surface melt. "*

You are completely right, that the calving event of the Amery Ice Shelf was long expected due to the developing rift as described by Fricker et al. 2002. Rift development was the key driver even though it remains unclear if the observed surface melt influenced the rift development during the second decade. We removed the misleading sentence and re-formulated the section:

*"The front of Amery Ice Shelf gradually advanced between 1997 and 2018. The environmental conditions changed with strengthening westerlies by + 0.26 m/s within the second decade and increased snow melt (+ 0.23 mm w. eq. per day) on the northern part of Amery ("Loose Tooth" region) over the last two decades but a decrease (- 0.1 mm w. eq. per day) in the southern part. The stability of the ice shelf is confirmed by velocity measurements where no speed-up occurred since the 1970s (Rignot et al., 2019). The basal melt rates for Amery Ice Shelf are low (Paolo et al., 2015) suggesting that the westerlies did not strengthen enough to cause upwelling CDW. In 2019, the tabular iceberg D-28 calved from Amery Ice Shelf. This calving front retreat was predicted by Fricker et al. (2002) based on the observed rift propagation and regular calving cycle of the ice shelf. Still, it remains unclear if the observed increase in surface melt influenced the rift propagation in the Loose Tooth region. Prior to 2006, the rift propagation of Amery Ice Shelf was not influenced by environmental forcing but the authors did not exclude the potential influence of surface melt if the mean air temperature would raise above zero (Bassis et al., 2008)."*

I suggest restructuring the paragraphs in the Discussion that focus on individual ice shelves or particular calving styles, to begin with mention of the complex interactions between the glaciological forcings and the environmental forcings that previous studies have identified to be involved in the calving events. Then follow this with the supporting evidence from the work performed here for the involvement of the environmental forcings in the observed frontal retreat. Including mention of the specific glaciological forcings as well as referencing studies that have looked at the glaciological drivers of calving will reduce confusion surrounding the key drivers, strengthen the arguments regarding the important influence of the environmental forcings and allow key environmental variables to be identified for future change analysis.

Thank you very much for mentioning the weaknesses in the discussion section and underlining misleading phrases. We completely revised the discussion section as suggested. First, we explain all involved factors (ice dynamics, external forcing, geometry) identified by previous studies and then provide supporting evidence from our work performed. This allows the reader to consider not only the environmental drivers assessed in this paper but also known glaciological factors that influenced the calving front retreat.

**Technical comments:**

Thank you very much for taking the time and proposing so many improvements regarding wording and gramma. We included all gramma/wording comments as suggested in the improved manuscript. Where necessary, we provide some additional information below:

Line 1. As only environmental drivers were evaluated with regard to calving front retreat I suggest the title should be amended to reflect this, e.g. 'Environmental Drivers of Circum-Antarctic...

Thank you for this idea. We changed the title of the manuscript accordingly to better differentiate from glaciological parameters and put the focus on environmental drivers.

Line 41. 'natural cycle of decay and growth'. This is a key point as it relates to the glaciological forcings that I mentioned in the Specific Comments. I suggest expanding what you mean by the decay and growth cycle and how glaciological parameters fit into

this. That way this paragraph can introduce the relationship between the glaciological parameters and the environmental forcings.

We welcome this comment and added a more comprehensive introduction to the relationship between glaciological parameters and environmental forcing. From L40 to L70 we included an explanation on factors influencing the calving front position including ice dynamics, geometry, external mechanical and external environmental forcing.

Line 113. 'sea ice months April through to October,' how have you chosen these months? Suggest adding clarification. Are you referring to fast ice or pack ice or both?

Those months were chosen in line with previous studies by Massom et al. 2013 and Miles et al. 2016 as now mentioned in the manuscript. The sea ice measurements cover fast and pack ice during those months.

Line 205. Best to avoid starting a sentence with 75%, suggest changing the sentence structure. Also I'm not sure what you mean by Ross West and Ross East, I don't think this is a common naming convention and I'm wondering if the Ross West is the McMurdo Ice Shelf? In addition, Figure 2 shows two Ross West labels.

We used the naming convention of the MEaSUREs Antarctic Boundaries for IPY 2007-2009 from Satellite Radar (Version 2) for all ice shelves. Ross East and West (we corrected the labels) originate from the border between the ice divides from EAIS and WAIS. This naming convention was used in previous studies as well e.g.

Rignot, E., S. Jacobs, J. Mouginot, und B. Scheuchl. „Ice-Shelf Melting Around Antarctica". Science 341, Nr. 6143 (2013): 266–70. https://doi.org/10.1126/science.1235798.

---

## Author Comment (AC2) · 22 Dec 2020

Response to Referee #2

As noted by reviewer Eleri Evans and in the comment by Brad Lipovsky, this study produces a very useful new Antarctic ice-margin dataset and mapping of frontal change which is a valuable addition to the literature. It also provides a useful and interesting look at patterns of change in some of the major ocean and climate parameters around Antarctica over the ERA5 period/last ~two decades, on a large scale and in a manner that is consistent and compelling. As the core results of this study, these are strong contributions (I suggest some minor improvements/clarifications below).

We would like to thank the anonymous referee for the constructive and comprehensive review of our manuscript. The mentioned improvements and clarifications are a welcome addition to our paper. Please find the answers on your comments below in blue color. The improved manuscript containing the described changes (highlighted with the track changes function) will be provided after we have received the final feedback from the editor.

The correlation analysis between the frontal changes and climate/ocean parameters is more challenging because the response of ice shelves and glacier fronts to forcing is so markedly nonlinear. Thresholds in the response to forcing are common, as are instabilities in which, by definition, the shelf/glacier behaviour becomes divorced from external forcing. These issues are alluded to in parts of the discussion, but need to be addressed.

We fully agree with you that the response of glacier and ice shelf fronts to forcing is often non-linear, even though a linear response of tidewater glaciers to ocean and atmosphere warming can exist (see Cowton et al. 2018). In contrast to the temporal correlation performed by Cowton et al. we assessed calving front change by a spatial correlation with environmental drivers. That means, we did not assess the linear response but the fact that an increase/decrease in assessed variable occurred with the retreat/advance of the calving front, hence the spatial relationship. This partly overcomes the difficulty with linearity as environmental changes and calving front response were averaged over one decade. But for sure, responses to forcings that occurred outside our observation period cannot be captured by our study design. We mentioned this in our added section about limitations of the study:

 "In addition, the natural calving cycle and responses to environmental forcing may prevail on longer time scales than the ones addressed in this study due to limited data coverage. For example, once ice shelf destabilization was initiated the response in frontal change might no longer linearly be connected to forcing and decouple from external forcings as known for the marine ice sheet instability (Feldmann and Levermann, 2015; Joughin et al., 2014; Robel et al., 2019)."

The behaviour of ice fronts could be seen as the combined result of:

i) Externally-forced trends like ice-shelf thinning due to increased basal melting (as in the Amundsen Sea Embayment), or the loss of the surface firn cover due to warmer summers (as on Larsen A and B). The forcing of these trends could potentially be diagnosed through correlation, if the right parameters can be measured for long enough (e.g., ocean temperature at depth, or positive summer surface air temperatures). Note though that the response to these forcings is not necessarily linear due to feedback. For example a reduction of albedo as a shelf surface melts acts as a positive feedback, enhancing the sensitivity of melt to shortwave radiation.

Positive feedback processes (e.g. albedo reduction due to surface melt) exist and are worth to be considered. In our case the effect of non-linearity due to feedback is minor as a spatial correlation was preformed (see description above).

ii) Externally-forced shocks superimposed on these trends, like an exceptionally warm summer (as on Larsen B). These would not be readily captured by decadal climate means and would not necessarily have the same effect on all shelves/fronts, so would be difficult to correlate to frontal change.

Threshold behaviour could be very important for i) and ii) – e.g., the difference between a summer surface temperature staying just below freezing or just above is profound, with the latter producing meltwater and rapidly densifying the firn. Rapid retreat or full shelf collapse could be triggered a slightly larger than normal retreat of a shelf front that happens to take it back behind a compressive arch of forces (e.g., Larsen A).

We fully agree that one limitation of our study design is the calculation of decadal means which by nature cannot account for extreme events and hence threshold behavior. This is very likely the cause why we could not find a significant correlation between relative increases in mean air temperature and glacier front retreat. This fact is now included in the discussion to avoid confusion about air temperature driven surface melt which indeed effects the stability of glaciers and ice shelves.

*"Relative changes in mean air temperature could not be identified as a direct driver for calving front retreat, even though increases of up to 2°C per decade were measured in some coastal areas (e.g. Dronning Maud Land, Victoria Land) which is beyond the uncertainty of the ERA5 air temperature data. This suggests that mean air temperature over a decade is not an appropriate way to assess the effect of air temperature on calving front change because the relationship between air temperature driven surface melt and hydrofracture is known to destabilize ice shelves and can cause glacier front retreat (Arthur et al., 2020; Banwell et al., 2013; Leeson et al., 2020a). More suitable would be the assessment of the amount of positive degree days and temperature extreme events directly influencing surface melt. For example, catchment wide melt and the resulting runoff had a higher impact on the retreat of Greenlandic glaciers than local air temperatures (Cowton et al., 2018). Howat et al. (2008) found that peak events in air temperature and sea surface temperature can initialize glacier retreat."*

*"A connection to mean air temperature changes could not be found because decadal average of climate variables can yield important information on long-term changes in environmental conditions but peak events were not captured. This already reflects one major limitation of our study. Especially regarding air temperature, threshold behavior (above zero degree days) and peak events are of major importance for the initialization of ice shelf destabilization."*

iii) Internal ice dynamics like the calving cycle, with a long, slow advance followed by an abrupt calving event, controlled by the evolving stress field and existing damage to the ice. This is largely unrelated to external forcing. For some shelves with long cycles, these may not be well sampled even by decades-long observations. Calving-cycle events can be large, dominating the statistics of frontal change.

We are completely aware that calving-cycles and forcing can prevail over longer time spans than assessed in this study. We mention this for example for Ronne and Ross Ice Shelf . This was already mentioned in the conclusion of the original manuscript but to further empathize this fact, we added an additional paragraph in the discussion.

iv) Unstable dynamic response to trends and shocks. Once initiated, thinning, acceleration, damage and retreat of shelves like PIG and Thwaites may run indefinitely, strongly controlled by evolving ice stress and damage, and only somewhat modulated by the sort of environmental parameters studied here. A marine ice sheet instability (MISI) could, for example, be triggered after several decades of ocean-driven shelf thinning and perhaps some shelf retreat, but once initiated could even drive an ice shelf readvance as ice dynamics took over, regardless of the external forcing. Similarly, a marine ice cliff instability (MICI) could be triggered by an initial external forcing but then progress into a runaway retreat regardless of what happens to that forcing.

The dynamic responses of ice shelves and glaciers to trends and shocks is very complex and might exceed the decadal observation periods of our study. To identify the initial start of forcing, longer time series of climate variables as well as calving front position would be necessary but unfortunately this data does not exist. Still, we wanted to address this fact in our manuscript to provide the reader a more holistic point of view and mention unstable dynamics in the discussion and added the following paragraph:

"*It has to be considered that the initial start of destabilisation in Pine Island Bay occurred previous to our observation period as mass loss exists since 1979 (Rignot et al., 2019), new rifting areas created in the beginning of the 1990s (Bindschadler, 2002; Rignot, 2002) and basal melt by ocean forcing exists at least since 1994 (Jacobs et al., 2011). Once marine ice sheet instability is initiated changes in ocean forcing can no longer directly be linked to ice flow and hence calving front position (Christianson et al., 2016). Consequently, it cannot be ruled out that the measured frontal retreat over the last two decades is actually a response to earlier destabilisation by ocean forcing. Nevertheless, our observations confirm that Pine Island and Thwaites Glacier are still exposed to ocean forcing.*"

I'd expect that these non-linearities, thresholds and instabilities mean that ice front changes are not likely to correlate well with external forcing….and yet…there are some signals there in the correlation results. And it does seem reasonable that a sustained forcing, like decreased sea ice cover, acting over a large area could drive a coherent signal of change at multiple independent ice fronts, particularly where their calving is dominated by frequent production of multiple smallbergs rather than rare, large tabular bergs. The forcings are important because they drive the trends and provide the shocks, but they might be difficult to untangle when looking at all of the ice fronts.

We agree that non-linearities, thresholds and instabilities exist which is why we decided to analyse frontal change over decadal time spans to account for longer-term forcing and evolving instabilities. For sure, it is difficult to explain all ice front changes because so many different factors influence the frontal position as you correctly stated above. Still, in many cases where we observed changes in environmental conditions glacier front retreat occurred as underpinned by the correlation analysis and the discussion.

**General suggestions**

Given the above, I suggest that instead of seeking to explain ALL ice-front changes through correlations with external forcing, the discussion section is re-oriented towards addressing the questions:

- i)   are ANY of these ice-front changes externally forced? (i.e., which ones can be distinguished with confidence from noise, internal dynamics and dynamic instabilities?)
- ii)   then for these fronts where forcing is detectable, which forcings have mattered most?

We appreciate your suggestion on re-structuring and re-orientation of the discussion. The introduction to the discussion is now more focused on the question if ice-front change is externally forced. Additionally, in Section 5.4 we provide information in which coastal sectors potential forcing by climatic variables was detected. Which forcing mattered most is difficult to identify as mostly a combination of internal and external forcing occurred and only detailed local studies will provide insights into this.

 "We want discuss if external environmental forcing was responsible for the observed glacier retreat or if internal glaciological forcing was the key driver."

To do this, perhaps choose subsets of the coast with numerous relatively small, independent ice fronts that are not experiencing major dynamic thinning, are not dominated by rare, major calving events at one or two shelves, and are not still responding to shelf collapses from several decades ago (like Wordie and various other AP coasts probably were). Look for correlations that are spatially coherent on the same scale as the forcing patterns, i.e., affect multiple neighbouring shelves/fronts simultaneously. Consider extracting statistics on decadally-extreme forcing events rather than just decadal means. Consider focusing on beyond-threshold parameters like summer-air-temperatures above-freezing (or positive degree days) rather than all temperatures. Consider calculating these temperatures only at very low altitude (e.g., <200 m or as appropriate) to focus on the shelves and ice fronts themselves - the 100 km landward buffer currently used will inevitably bias the shelf/front temperatures low, and this could be important. The bias will be particularly big for small, fringing shelves with relatively steep ice sheet inland. While this involves some extra analysis, I think that you already have the datasets to focus in on these questions.

Thank you very much for proposing additional analysis to improve our analysis. Unfortunately, this is not as straight forward as it might first look like.  Selecting just a subset of the coastline would reduce the input parameters for the correlation drastically. For example, a coastal section where simultaneous terminus retreat occurred is Wilkes Land consisting of eight glacial basins. This would be a very small number of parameters going into the correlation analysis. Regarding the spatially coherent correlations on the same scale as the forcing pattern, the circum-Antarctic correlation exactly yields this information through the spatial correlation. The strength of the circum-Antarctic analysis is the amount of glacier basins going into the analysis and that for each glacier basin it is evaluated if advance/retreat occurred along with changes in climate variables.

We agree that statistics on decadal extreme forcing events would be interesting and would yield very interesting data on peak events that are especially interesting regarding air temperature. Unfortunately, the calculation of decadal means is based on monthly mean data which does not include information on extreme events or positive degree days. To account for that we stress this shortcoming of our analysis:

"A connection to mean air temperature changes could not be found because decadal average of climate variables can yield important information on long-term changes in environmental conditions but peak events were not captured. This already reflects one major limitation of our study. Especially regarding air temperature, threshold behavior (above zero degree days) and peak events are of major importance for the initialization of ice shelf destabilization."

Your concerns of the 100 km landward buffer are reasonable even though an elevation restricted approach would also involve some difficulties. We tested your suggestion by masking the Antarctic Ice Sheet by a 200 m elevation threshold. The area of big ice shelves like Ronne and Ross would then extend far land inward (in some cases beyond the grounding line). For very small fringing shelves only one to three pixels of temperature data would remain for calculating temperature change with would create high uncertainties due to the low amount of data points. We found that the 100 km buffer is the best trade-off between including most off the shelf area and not including too much of the ice sheet area for a circum-Antarctic analysis.

**Specific suggestions/questions**

Section 3.1 on coastline detection –physically what aspects of the HH/HV signal distinguish the 'ocean' and 'land' classes? i.e., why is it desirable to have HV as well as HH? A contrast in volume-scatter from the land ice and sea ice?

Different polarizations make it easier to distinguish between different ice types and ocean. HH-and HV-polarizations are used for ice type and ice edge detection. The HH polarization is suited for water-ice discrimination and the contrast between smooth and rough ice is good. HV polarization is less sensitive to a wind roughened ocean and has low backscatter values over the open ocean. Additionally, the cross-polarized signal is more influenced by rough features (e.g. ice) than a like-polarized signal. Hence, the cross-polarized signal varies more depending on ice type compared to the co-polarized signal. The volume scatter of sea ice varies depending on the age of the ice (older ice has more volume scattering). Glacier ice has higher surface scatter, except after fresh snowfall where volume scatter appears.

For reference:

Dierking, Wolfgang, und Leif Toudal Pedersen. „Monitoring sea ice using Envisat ASAR - A new era starting 10 years ago". In *2012 IEEE International Geoscience and Remote Sensing Symposium*, 1852–55. Munich, Germany: IEEE, 2012. https://doi.org/10.1109/IGARSS.2012.6351147.

Ressel, Rudolf, Anja Frost, und Susanne Lehner. „Comparing automated sea ice classification on single-pol and dual-pol terrasar-x data". In *Geoscience and Remote Sensing Symposium (IGARSS), 2015 IEEE International*, 3442–45, 2015.

Partington, Kim C, J Dominic Flach, David Barber, Dustin Isleifson, Peter J Meadows, und Paul Verlaan. „Dual-Polarization C-Band Radar Observations of Sea Ice in the Amundsen Gulf".

*IEEE Transactions on Geoscience and Remote Sensing* 48, Nr. 6 (Juni 2010): 2685–91. https://doi.org/10.1109/TGRS.2009.2039577.

Park, Jeong-Won, Anton Andreevich Korosov, Mohamed Babiker, Joong-Sun Won, Morten Wergeland Hansen, und Hyun-Cheol Kim. „Classification of Sea Ice Types in Sentinel-1 Synthetic Aperture Radar Images". *The Cryosphere* 14, Nr. 8 (20. August 2020): 2629–45. https://doi.org/10.5194/tc-14-2629-2020.

What decisions did you make in defining messy fronts like the collapsing Thwaites Glacier/Iceberg tongue?

We defined all fronts after the delineation procedure explained in Baumhoer et al. 2019*: "In order to create accurate and consistent front labels, we define the calving front as the border between ocean and land ice including floating ice shelves and glacier tongues. As soon as an iceberg calves and is no longer connected to the ice shelf or glacier, it is considered as ocean."*

Line 142: why use winter scenes rather than summer when open water is more likely?
Summer scenes often include surface melt on the glacier. This reduces the backscatter and makes it often impossible to differentiate between open ocean and glacier ice. During winter this is not the case.

Section 3.1: can you give more detail on the uncertainty assessment? The total uncertainties of ±29 and ±144 sq km given in the abstract seem exceptionally small.

Line 172: do these uncertainties have a sign or are they ±? Is there a tendency towards biasing the fronts too far seaward, because sea ice/melange is sometimes present on the seaward side, and sometimes mistaken for 'land ice' (but not the other way around)?

We would like to address both comments above together as they relate to the same topic. You are completely right that the previously calculated uncertainties were too small as our initial uncertainty calculation could not account for different uncertainties in retreat and advance (see Table 2: uncertainties are the same for retreat and advance). Hence, the retreat and advance compensated each other resulting in small uncertainty values.

The coastline products included seaward/ landward biases but through manual corrections in areas of fast ice/sea ice/mélange regions the bias was reduced to a minimum. The small remaining biasing exists due to different spatial resolutions of the satellite imagery mosaics (MODIS vs. Radarsat and Sentinel-1) and over very steep rock slopes due to differences in SAR imagery vs. optical imagery which can lead to positive/negative uncertainties why we use the ± sign. To account for this biasing and improve our uncertainty assessment we created a better calculation approach where uncertainties are separately assessed for retreat (landward uncertainty) and advance (seaward uncertainty) not allowing to compensate each other anymore (even though this likely occurs). For the total areal changes of -29618 km$^2$ the newly calculated uncertainty of ±1193 km$^2$ is the mean of the uncertainty of retreat (± 1286 km$^2$) and advance (± 1100 km$^2$) for entire Antarctica. The uncertainties calculated with the new approach were updated throughout the manuscript and in Table 2.

Line 167: how did you define the 30 'stable' areas used for quality control?
We randomly selected 30 areas over ice and rock coastline where no frontal change exists (identified by high-resolution optical satellite imagery). The areas are distributed around the Antarctic coastline to cover all different kinds of stable rock coastlines from different angles.

Line 192: what do these '% of total area' mean? i.e., the 'total area' of what? Do you mean 'total coastline length' instead of area?

We meant % of the total changed area (now mentioned in the script). For example, within the first decade 24006 km$^2$ advanced (31 %) and 53624 km$^2$ retreated (69 %) creating a total changed area of 77630 km$^2$. This shows the share between retreat and advance as visualized for the individual ice shelves with pie charts in Figure 2.

Line 231: Do you mean "by 1°C" rather than "of 1°C"?

Changed as suggested.

Figure 6: Larsen B is showing up as having a big increase in snowmelt, but in fact it had by then collapsed.

You are completely right, we also recognized this error in the snow melt data and also criticized the inaccuracies of the ERA5 snowmelt data especially at the margins of ice shelves. In case of Larsen B we hypotheses that for the modelling an old coastline product was used where the Larsen B ice shelf still existed and hence melt was modelled over this area even though the shelf did not exist anymore.

*"Moreover, the low resolution of melt data did not always match the actual calving front. We assume that the use of a not up to date coastline product (e.g. at Larsen B) caused the inaccuracies over the ice shelf margins."*

Line 299: what does "the percentage of retreat/advance within each glacier/ice shelf basin" mean?

The proportional share between retreat and advance is given in % as shown in Figure 2 by the pie charts. This is also described in Section 3.3 from L 183 in the originally submitted manuscript: *"To remove the effect of different basin sizes and different amounts of ice discharge we took the percentage of advance and retreat within each basin instead of the absolute value for the correlation"*

Line 376: in contrast, Larsen C has not broken up. It did have a big calving event though.

Thank you for drawing attention to this misleading wording. Changed accordingly in the manuscript.

---

## Author Response (AR1)

Dear Editor,

thank you very much for your helpful comments.

We added a data availability section were the access to the extracted coastline and other used data sources is provided. The coastline will be available through the EOC Geoservice of the German Aerospace Center. As soon as the coastline is integrated in the Geoservice we will add the link in the manuscript before acceptance. The change rates are provided in the supplementary files as it is more suitable to offer them in a table than at the Geoservice.

Regarding your specific comments:

1. We revised the entire manuscript to provide more precise wording and to better distinguish between different kinds of driving forces (e.g. L41 ff., L654 ff.). Moreover, we stress the limitations of our diagnosis especially regarding threshold behaviour (e.g. L375 ff., L648 ff.).
2. In the revised version, we stress the importance of extreme events at several sections of the manuscript. Please see L21, L618, L668.
3. We included the statement in section 3.3. beginning from line 217.

We hope our revised manuscript meets your expectations.

Kind regards,

Celia Baumhoer (in behalf of all co-authors)

---

## Referee Report (RR1)

1020

[referee-annotated manuscript omitted]

---

## Author Response (AR2)

**Response to Editor Comments**

We would like to thank you, as an editor, and also Eleri Evans for taking the time to review our manuscript again. Your comments and suggestions are very valuable and improved the quality of the manuscript. We are glad to hear our efforts in the revision process are appreciated.

All editorial comments were included additionally to the very detailed comments of Eleri Evans (we very much appreciate your efforts!). Please find more detailed answers for your additional concerns below in blue color. Page and line refer to the revised track changes document.

Your original manuscript was reviewed by two referees, and the revised one was reviewed by one referee. These referees and I concerned to identify driving forces from regional correspondences between coastline migration and environmental factors. You made excellent work to address our concerns. This point is no longer my concern. Also, clarify and readability of the manuscript were largely improved. So, first of all, I would like to thank authors' effort in this revision process. I have several more concerns, and many editorial suggestions. The referee also provided very thoughtful comments in pdf; these are mostly editorial, but suggest local restructuring from sentences to a paragraph level at several locations. Therefore, I request a minor revision. Please provide the track-change manuscript and a response letter when you submit the revised manuscript. No responses are necessary for purely editorial issues, but I would like to see responses to following relatively major issues. In the comments below, all page and line numbers refer the track-change manuscript.

1. I am somewhat confused with "ice shelf and glacier front" that appears so many times in the manuscript. P2L31 refers Nicholls et al. (2009) and says "ice shelves and glaciers are the floating extensions of the ice sheet". I cannot find this definition in that reference; if I am right, please delete this sentence/reference. Also, in my opinion, this definition is wrong. The glaciers are grounded. The term "glacier tongues" are sometimes used to refer fast-flowing part of an ice shelf that is fed by a fast-flowing glacier. However, for your work, I don't see a specific need to distinguish glacier tongues and other ice shelves. About one quarter of Antarctic coastline has no ice shelves and the ice sheet is terminated with grounded ice. Strictly speaking, the calving front does not refer most of these grounded-terminated margin. Please add more clarification on this issue and define terms used in this paper in Introduction. The manuscript is clear enough overall so that I don't see a strong need to change the terms entirely in this paper, but some clarification is necessary. You do not need to provide a long argument about terminology, but please make your own judgement which terms are most appropriate and provide a short summary in the response letter.

Thank you for mentioning this concern. We checked again on Nicholls et al. (2009). With our definition we referred to the following section in Nicholls et al. (2009), p.3:

"Ice shelves form when the ice at an ice sheet's oceanic
boundary does not calve as icebergs at the point where it goes
afloat (the grounding line) but remains connected to the
grounded ice sheet. Ice shelves may thus be regarded as
floating extensions of the ice sheet, and whether an ice shelf
forms depends to a large extent on the coastal geometry"

But you are completely right, this definition relates to floating ice shelves and not grounded glacier termini. To make our definition clearer we changed the manuscript accordingly and shifted the definition to P2L43 for a better reading flow: *"The coastline of the Antarctic Ice Sheet is defined as the border between the ice sheet and the ocean (Liu and Jezek, 2004), extending along glacier and ice shelf*

*fronts. Throughout this paper, we refer to floating glacier tongues and ice shelves when using the term "glacier and ice shelf front" as well as "calving front"* and removed the definition of coastline at P5L111 to avoid repetition. Additionally, we address grounded and floating termini in the coastal change analysis more clearly: "*Directly calculating coastal change between these coastline products includes changes in floating calving fronts and grounded ice walls (as shown in Figure 1) even though the amount of change from grounded termini is small and prone to inaccuracies as limited due to the spatial resolution of the coastline product from 2009.*" P5L122.

We hope this addition helps with clarification and emphasizes our focus on floating glacier tongues and not grounded glacier termini. If you still think this definition needs changes, please don't hesitate to contact us.

2. It is often said "front retreat", but more accurately I think the authors generally refer "front position movement", including both retreat and advance (e.g. P1L12).

Thank you very much for this comment. We checked our manuscript and changed the term "front retreat" to "front movement, fluctuation or change" where appropriate.

3. Most of geographical names mentioned in the paper are shown in Figs. 1 and 2, but not all. I commented below as I see this issue; please carefully review the manuscript and make sure that the all names are labeled in these figures. It is also helpful if relatively not-well-known names are assocaited with the name of regions, such as "xx glacier in Dronning Mauld Land" using the regional name shown in Fig. 1. At an appropriate location (end of Introduction, or first time when a geographical name is mentioned in the paper), please mention that all geographical names are shown in Figs. 1 or 2.

The geographical names are now all mentioned in Figure 1 or 2 (e.g. Nickerson, Larsen G Ice Shelf). Additionally, we refer to both Figures when introducing a new name so the reader can locate it in the map (e.g. in the discussion section).

4. Snowmelt is shown with the unit of mm w eq. per day. Why is this unit chosen? Snowmelt is examined over three months, December, January, and February, in 90 days in total (P7top). Is it better to show snowmelt in the unit of "mm water equivalent per year" (i.e. current value times 90)?

Changing the unit to year might be appropriate in some cases. In ours, there are two reasons why we chose melt per days instead of melt per year. ERA5 monthly means snowmelt data has the unit m w eq. per day which is why we decided to keep the unit "day" instead of "year". Additionally, most of the melt occurs within the selected months Dec, Jan and Feb but in some years, melt might occur beyond that time frame. Even though we could change the temporal value to year it would suggest we calculated the snowmelt mean for the entire year (and not only the summer months). To avoid this potentially misleading assumption we would like to keep the unit of days.

5. Section 4.8 present correlations between climate variables and calving front position movement. Are both 1997-2008 and 2009-2018 period data used together? And then why are the reference periods shown separately? This point should be better clarified in the text.

You are completely right, we used the data of both decades (1997-2008 and 2009-2018) together for the correlation. We explain this in 3.3 P9L240: "This created 14 different variables which were correlated with each other based on 188 observations (N=188). The number of observations is derived

from 94 assessed glacier basins with variable averages for the two different decades (1997-2008 and 2009-2018)."

The reference period (1982-1996) was not directly used for the correlation, as during this time frame no information on the calving front position is available. To still use this data as an indicator how the environmental conditions changed between the first/second decade and the reference time period we used the label "relative". We show both, relative and absolute values as both indicate different conditions. Relative values indicate that a front retreated/advanced in conjunction with a variable that is higher/lower compared to the reference time period. In contrast, the absolute values just indicate that a front retreated/advanced in conjunction with high/low values of a specific variable. We added a sentence and the text reads as follows: "To also assess relative changes in the variables to previous times we subtracted the reference mean (1982-1996). This means the relative values indicate the change of an environmental variable within the first/second decade compared to the reference time frame" (P9L229).

**Minor issues.**

We edited all mentioned minor issues in the revised manuscript. In some cases, additional explanations were necessary which we mention in blue color below.

P1L23: remove CDW (you don't need to define it in Abstract).
P2L55-56: maybe better to explicitly mention ice rumples (pinning points), though it is included in bed topography in general.
P3L91: add "in Antarctic Peninsula" after Wordie Ice Shelf. Wordie is shown in Fig. 2 but still some geographical guide is very useful for readers.
P6L121: do not cite the dataset as NSIDC 1997 or such (there are several more cases when datasets are cited).
P6L126: change the title to "ERA 5 reanalysis data"
P7L148: change to "lateral resolution"?
P8L182: methods (plural)
P9L224: confusing. Better to say "for the three time periods 1982 – 1996, 1997 – 2008, and 2009-2018. The first period is used as the reference period to measure temporal changes in the following two decades."
P11L255: change to West Antarctica. (I like your style not to use AP, EAIS, and WAIS in the main text, which is much easier to read than the other style so please keep it more consistent).
P11L259: confusing sentence. Maybe "Excluding these two ice shelves, the rest of West Antarctica …"
P11L261 and elsewhere: I am not an English native speaker but "within the first decade" sounds like you have data to show variability within the decade.
P13L291: cite Fig. 4 beginning of Section 4.3, not very end.
P14: change EAIS and WAIS to East Antarctica and West Antarctica at all locations in this section.
P14L310: Add Celsius at several locations(here -0.5oC and at the bottom of the page +0.25oC). Also change kelvin to oC here and at other places.
P15L32: cite Fig. 6 at the beginning of this section.
P15L334: cite Fig. 7 at the beginning of this section.
P17L354: Figure 8 (not 9).
P18LL365, 366, and 368ff: This is a good example. It is said "retreat" but is it better to say "position"?
P18L375ff: do not define r_summer or such, and write it like "(r = 0.18 for summer and r=0.23 for winter)"
P20L408: is it "exceptionally" positive? Better to be a bit more specific. The other two larger peaks are mentioned at another part of the paper, and now smaller peaks are referred as exceptionally positive.

P20L415: change to 1998/1999.
P20L417 and 419: Wordie Bay and Marguerite Bay are referred. Is it possible to call these bays using ice shelf names in Fig. 3? If you decide to refer these bays, add them to Fig. 2.
P20L427: typo? Surface melt instead of basal melt
P21L445: change to "increase surface melt"
P21L450-451: better to say "it remains unclear whether…."
P21L463: what is Larsen D-G?

P22L472: change to West Antarctica (Section 5.2 header)
P22L476: here and at many other places it is said "an entire area of xx". Do you need to say "entire area" at these locations? Is it clear enough if you say "an area of xx"?
P22L476: did you estimate the iceberg areas using BYU dataset? Then clarify the data source.
P22L479 and 480: change to A38 and A39, and A43 and A44.
P22L487: Did you measure it over the calving area or over the ice shelf including the calving area? Confusing.
P22L487 and 488: Here it is said Ronne-Filchner Ice Shelf, but it is refereed as Ronne Ice Shelf at other locations. Please keep it consistent.
P22L496: change "reference mean" to "reference period".
P23L522 Nickerson Ice shelf does not appear in Fig. 3
P23L526 change to East Antarctica
P23 bottom: cite Fig. 2
P24L543: typo? "in sea ice no wind direction"?
P24L544 and 545: change to "up to -1oC" and "up to -0.49oC" to keep this paper consistent.
P24L554: typo, CDW.
P24L561: can you be more specific than "more recent calving event"?
P25L568: change 1982-1996 to the reference period.

P25L578: change northern and southern to seaward and landward.
Unfortunately, landward and seaward is not appropriate for this geographical setting. We added the Mackenzie Bay (also added to Figure 1) for describing the location better.

P25L584: who is the authors? Revise this sentence for clarity.
P25L595ff: calving area along the Dronning Maud Land was analyzed in this paper, Goel et al. : Characteristics of ice rises and ice rumples in Dronning Maud Land and Enderby Land, Antarctica, J. Glaciol., doi: 10.1017/jog.2020.77, 2020. 1-15, 2020.
P25bottom: change +0.9 – 1.8oC
P26L619: change "measured changes" to "measured increases"?
P27L654 and 660: do you use "break-up events" to refer only tabular iceberg calving? If they are used to distinguish infrequent large calving events other more frequent calving events, it is fine. However, if otherwise, please revise them to calving events.

P28L684: add data source after "natural calving cycle"
We decided to re-phrase the sentence for a clearer understanding. The conclusion on the non-exceptional calving of the Ross and Ronne ice shelves is explained in Section 5.2 (P22L490ff) so we thought it would not be necessary to mention the references again as this would be unusual in the conclusion section. If you still wish for citing the references again we will include the references (Ferrigno et al., 2007; MacAyeal et al., 2001; Ferrigno et al., 2005).

P29L707: typo, CDW

**Figures:**

Figure 1: Add reference to LIMA mosaic. I suggest that Figure 1 shows regional names and non-ice-shelf names such as bays, whereas Fig. 2 shows all ice-shelf names. For example, it is unclear whether the label "Wordie" refers Wordie Bay or Wordie Ice Shelf (Fig. 3 also shows Wordie). Also "Shirase" appears both in Figs. 1 and 2, but it is said "Shirase Bay" in Fig. 1 and Shirase in Fig. 2 so it is clearer. Also, please consider whether bays are needed to refer in this paper.

We added the link to the LIMA Mosaic in the data availability section. To better distinguish between different features (ice shelves, bays, seas etc.) we used different colors and describe the color-code in the figure caption.

Figure 2: Add the boundary of EP-F and E-EP over the Ross Ice Shelf. Similarly add the boundary over the Ronne-Filchner Ice Shelf. This is an important boundary to distinguish EAIS and WAIS. To show the boundaries in the Peninsula more clearly, is it possible to add a zoom-up panel showing Peninsula?

Thank you for your advice. The boundaries between EAIS and WAIS were added to Figure 2 as well as a zoom panel for the Antarctic Peninsula. Additionally, Nickerson Ice Shelf and the remaining Larsen ice shelves are mentioned.

Figure 3: re-consider the colorbar used for the absolute plot. This colorbar is suitable to show positive/negative anomalies but not good to show absolute values. In the caption add "(1982-1996)" after "the reference mean" to improve clarify (you don't need to do it every time but it is the first time to show the reference mean/period).

The color bar for the absolute temperature was updated to an increasing color scheme.

Figs. 4, 5, and 7 do not show the reference period (probably for space saving purpose). However, I think another panel showing the reference period is very useful, particularly for Fig. 7 because general wind direction is useful when "weaken easterlies" or "strengthen westerlies" are said. If space is still your concern, consider adding the reference panel as the supplement.

We decided to add the absolute values for those figures to the supplement for space saving purposes. Please have a look at the edited supplementary file.

Fig. 6: change the top right panel title to "Reference" (now it is said "long-term"). If you keep this unit, clarify in the caption that snowmelt is measured over 90 days from December to February.

We added the term "reference" and mentioned the calculation over the summer months in the caption.

Fig. 7: do not define acronym in the caption. Remove CDW from here and define CDW in the main text.

Done.

Figure 8: use background color to show the reference and following two decades. The dash bars do not work well for this purpose. Also, update the reference for SAM; the data are shown to 2018 here, but the data source was published in 2003. Is this correct?

Thank you for this hint. We cited the method paper on the SAM Index calculation but now also added the website where the data can be downloaded. Additionally, instead of the dashed lines we use color panels for a better visualization of the different periods.

Table 2: the second table top left cell shows "total". Typo? Maybe "basin" or just keep it empty.

Thank you for this comment. "Total" was related to entire retreat and advance in contrast to the annual values above. But you are completely right that this is misleading at this point. We changed "total" to "region" to be consistent with the table above.

---

## Author Response (AR3)

Dear Editor,

thank you very much for editing our manuscript. We are delighted to hear that the manuscript is finally accepted. The data set citations were updated as indicated and the typos removed. Additionally, we corrected the supplementary material.

Thank you again for your helpful comments and patience in the review process.

Kind regards,

Celia Baumhoer (in-behalf of all co-authors)